

# Approach to the separatrix with eccentric orbits

**Guillaume Lhost[1]⋆ and Geoffrey Compère[2]†**

**1** Physique de l'Univers, Champs et Gravitation, Université de Mons – UMONS,
Place du Parc 20, 7000 Mons, Belgium
**2** Université Libre de Bruxelles – International Solvay Institutes – BLU-ULB space center,
CP 231, B-1050 Brussels, Belgium

⋆ guillaume.lhost@umons.ac.be , † geoffrey.compere@ulb.be

## Abstract

Eccentric binary compact mergers are prime targets of current and future gravitational wave observatories. In the small mass ratio expansion, post-adiabatic inspirals have been modeled up to the separatrix, where first-principle modeling currently ends. In this paper, we derive the analytic late time solution to the adiabatic inspiral in terms of self-force coefficients at the separatrix. We identify the role of the Lambert $W_{-1}$ function as a key mathematical ingredient in the approach to the separatrix.

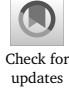
# 1   Introduction

Solving the two-body problem in General Relativity has never been as pressing as today, given the breakthrough of gravitational wave observations of compact binary mergers [1] and prospect of upcoming ET, LISA and Cosmic Explorer observatories. One method to address the two-body problem is the small mass expansion or self-force framework [2]. Based on the mass ratio to separate timescales [3], adiabatic waveforms have now been numerically generated during the inspiral phase for generic orbits [4], which is the first step towards the much larger program to model with a numerically efficient scheme the full parameter space of post-adiabatic generic inspiral-to-plunge systems including, in particular, eccentricity.

Eccentric orbits have been modeled within the self-force framework [5–7]. Numerical evaluations of the self-force quantities have been performed [8–10] and orbits have been numerically generated with increasingly faster evaluation schemes [11–19]. Only the inspiral motion has been computed so far from first principles within the self-force formalism in the eccentric case, though several hybrid or approximation schemes have been formulated [20–27]. In the mass ratio expansion, adiabatic and post-adiabatic inspirals become ill-defined at the separatrix (for a review, see [2] and for a thorough description of the separatrix, see [28]). Formulating a consistent dynamics around the separatrix is challenging. For the non-eccentric case, the motion around the innermost stable circular orbit has been understood in a consistent expansion scheme [29–31] following the seminal work of [32–34].

In this paper, we aim to get one step closer to the analytic understanding of the motion close to the separatrix, in order to build a first-principle scheme for a transition-to-plunge expansion which would generalize the quasi-circular case [29,30]. The main aim of this paper is to analytically solve for the late time dynamics of the adiabatic inspiral with eccentricity in terms of self-force coefficients at the separatrix. Along the way, we will reproduce and extend former analytical work for this system [5,6].

The rest of the paper is organized as follows. We first review the description of eccentric orbits around the Schwarzschild black hole in Section 2. We develop the analytic solution close to the separatrix in Section 3. We conclude in Section 4. Technical expressions are relegated to the appendices. The notebooks with a summary of technical expressions as well as partial proofs is provided on this Github repository for the ease of reproductibility.

# 2 Review of eccentric orbits around Schwarzschild

In this paper, the main objective is to derive the analytic solutions of the adiabatic equations of motion of the eccentric equatorial inspiral asymptotically reaching the separatrix in a Schwarzschild background. Before starting the involved computation of the corrections due to the mass of the secondary, we set our notation by first reviewing the motion of a test-particle that spans a bound orbit around a massive primary black hole. In this section, we recall the geodesic equations around a Schwarzschild black hole located at the center of the coordinate system. The conserved quantities and the Kepler-like parametrization are discussed. We use the conventions of [2]. From now on, we set the spin of the primary to zero and the geodesic motion to be equatorial.

## 2.1 Geodesic equations

Consider a primary massive spherically symmetric compact object at the center of the dimensionless Boyer-Lindquist coordinates $\{t, r, \theta, \phi\}$. The spacetime metric of such system is given, by the Schwarzschild metric

$$ds^2 = M^2\left(-\left(1-\frac{2}{r}\right)dt^2 + \left(1-\frac{2}{r}\right)^{-1}dr^2 + r^2 d\theta^2 + r^2\sin^2\theta\,d\phi^2\right), \tag{1}$$

where $M$ is the mass of the primary compact object in geometric units. The standard dimensionful Boyer-Linquist coordinates are $\{Mt, Mr, \theta, \phi\}$. A test-body that moves in this metric is located at $z_p^\mu = (t, r, \theta, \phi)$ and has a velocity given by the quadrivector $u^\mu = (\dot{t}, \dot{r}, \dot{\theta}, \dot{\phi})$ where the operator "$\cdot$" denotes a derivative with respect to the (dimensionless) proper time per unit primary mass $\tau$. We denote the conserved dimensionless specific energy $E := -M^{-2}u_t$ and the dimensionless specific azimuthal angular momentum as $L = M^{-2}u_\phi$. We use the primary mass to adimensionalize all quantities. We set the Carter constant to zero to describe the equatorial motion. The geodesic equations, together with the mass-shell constraint $u^\alpha g_{\alpha\beta}u^\beta = -1$, are given by

$$r^4\left(\frac{dr}{d\tau}\right)^2 = R(r) := E^2 r^4 - (L^2 + r^2)(r^2 - 2r),$$
$$r^2\frac{dt}{d\tau} = f_t^{\text{geo}}(r) := \frac{r^2 E}{1-\frac{2}{r}}, \tag{2}$$
$$r^2\frac{d\phi}{d\tau} = f_\phi^{\text{geo}} := L.$$

The quantities $f_t^{\text{geo}}(r)$, $f_\phi^{\text{geo}}$ are the "instantaneous frequencies of motion", which we denote with the letter $f$ as in the review [2].

## 2.2 Quasi-Keplerian parametrization of bound orbits

We only consider bound orbits that oscillate between the periapsis $r_p$ and the apoapsis $r_a$. The (dimensionless) semi-latus rectum $p$ and eccentricity $e \in [0; 1[$ are defined as

$$r_p := \frac{p}{1+e}, \qquad r_a := \frac{p}{1-e}. \tag{3}$$

The radial potential can be reexpressed in terms of these roots as

$$R(r) = (E^2 - 1)\,r\,(r - r_p)(r - r_a)(r - r_3), \tag{4}$$

where the third root is

$$r_3 = \frac{2p}{p-4}. \tag{5}$$

The specific energy and angular momentum can be written as functions of $e$ and $p$ as [5]

$$E^2 = \frac{(p-2)^2 - 4e^2}{p(p-3-e^2)}, \qquad L^2 = \frac{p^2}{p-3-e^2}. \tag{6}$$

Bound orbits exist above the separatrix characterised by $p = 6 + 2e$. At this specific point, $E$ and $L$ take the values

$$E_\star(e) = \frac{2\sqrt{2}}{\sqrt{9-e^2}}, \tag{7}$$

$$L_\star(e) = \frac{(6+2e)}{\sqrt{3+2e-e^2}}. \tag{8}$$

In this paper, the subscript $\star$ given to a quantity $X$ denotes that $X$ is evaluated at the separatrix. Whenever $p > 6 + 2e$, that is $r_3 < r_p \le r_a$, the test body's radius oscillates between $r_a$ and $r_p$. This fast motion, in complement with the fast azimuthal motion, can be parametrized by the relativistic anomaly $\psi_r$ defined by

$$r(\psi_r) = \frac{p}{1 + e\cos(\psi_r)}. \tag{9}$$

Thus the geodesic motion for $r$ is equivalent to the geodesic equation for $\psi_r$ that reads

$$r^2 \frac{d\psi_r}{d\tau} = \frac{p^2 \sqrt{p - 6 - 2e\cos(\psi_r)}}{\sqrt{p^3(p-3-e^2)}} =: f_r^{\text{geo}}(\psi_r). \tag{10}$$

The advantage of this parametrization is that the growth of $\psi_r(\tau)$ is monotonous, which allows for a simple numerical solution starting from an initial $\psi_r^0$. This is an improvement over the evolution of the radius, which has turning points where $\dot{r} = 0$. The time and azimuthal phases are then solved as

$$\phi(\tau) = \phi_0 + L \int_{\psi_r^0}^{\psi_r(\tau)} \frac{d\psi_r'}{f_r^{\text{geo}}(\psi_r')}, \tag{11}$$

$$t(\tau) = t_0 + \int_{\psi_r^0}^{\psi_r(\tau)} d\psi_r' \frac{f_t^{\text{geo}}[r(\psi_r')]}{f_r^{\text{geo}}(\psi_r')}. \tag{12}$$

## 2.3 Fundamental Boyer-Lindquist frequencies

Since the motion of interest is periodic, we can use the action-angle variables as phase space variables to describe the motion. Here, we denote by $\varphi_\alpha$ the set of angle variables $\{\varphi_t = t, \varphi_\phi, \varphi_r\}$. We warn the reader that Greek indices label both spacetime components and components of the angle variables, depending upon the variable being used, as should be clear from the context. They are linear in time, and their linear growth is dictated as

$$\frac{d\varphi_\alpha}{dt} = \Omega_\alpha, \tag{13}$$

where $\Omega_\alpha =: \frac{\Upsilon_\alpha}{\Upsilon_t}$ are the dimensionless Boyer-Lindquist fundamental frequencies expressed in terms of the fundamental Mino frequencies $\Upsilon_\alpha := \lim_{\Lambda\to\infty} \frac{1}{2\Lambda} \int_{-\Lambda}^{\Lambda} d\lambda f_\alpha^{\text{geo}}$ where Mino time $\lambda$ is related to proper time $\tau$ using $d\lambda = r^{-2} d\tau$. The explicit analytic form of these functions can be found in the Black Hole Perturbation Toolkit (BHPT) following [35]. They are pure functions of eccentricity and semi-latus rectum; in particular $\Omega_t = 1$.

In terms of the angle variables, any function $\xi[r(t)]$ can be expanded as a Fourier series

$$\xi[r(t)] = \sum_{k \in \mathbb{Z}} \xi_{(k)} \, e^{-ik\varphi_r(t)} \,. \tag{14}$$

For a generic non-resonant orbit, $\xi_{(0)}$ is the average of $\xi$ over a period which we also denote as $\xi_{(0)} := \langle \xi \rangle$. The generic mode $\xi_{(k)}$ can be written as an integral over the phase $\psi_r$ (see e.g. [2])

$$\xi_{(k)} = \langle \xi[r(\psi_r)]e^{ik\varphi_r(\psi_r)} \rangle := \frac{\Omega_r}{2\pi} \int_0^{2\pi} \frac{f_t^{\text{geo}}[r(\psi_r)]}{f_r^{\text{geo}}(\psi_r)} e^{ik\varphi_r(\psi_r)} \xi[r(\psi_r)] \, d\psi_r \,, \tag{15}$$

after using Eq. (9). The function $\varphi_r(\psi_r)$ is known analytically in terms of elliptic integrals after solving for $\frac{d\varphi_r}{d\psi_r} = f_t^{\text{geo}} \Omega_r / f_r^{\text{geo}}$. We can therefore rewrite the integral as

$$\xi_{(k)} = \langle \xi[r(\psi_r)]e^{ik\varphi_r(\psi_r)} \rangle := \frac{1}{2\pi} \int_0^{2\pi} e^{ik\varphi_r(\psi_r)} \xi[r(\psi_r)] \, d\varphi_r \,. \tag{16}$$

As a direct consequence of these definitions, we have $\langle \frac{d\psi_r}{dt} \rangle = \Omega_r$, $\langle \frac{d\phi}{dt} \rangle = \frac{\Omega_r}{2\pi} \oint d\phi = \Omega_\phi$ where the last integral is over an orbital period.

We can formally express the solution to the relativistic anomaly $\psi_r$ in the geodesic case in terms of the angle variable $\varphi_r$ as

$$\psi_r^{\text{geo}}(\varphi_r) = \varphi_r + \Delta\psi_r(\varphi_r) - \Delta\psi_r(0) \,, \tag{17}$$

where

$$\Delta\psi_r(\varphi_r) = \sum_{k \neq 0} \frac{-1}{ik\Omega_r} \left( \frac{d\psi_r}{dt} \right)_{(k)} e^{-ik\varphi_r} \,. \tag{18}$$

We did not derive the analytic expression for the Fourier coefficients $\left( \frac{d\psi_r}{dt} \right)_{(k)}$, see [36] for relevant recent work. Instead, the Mino angle variables $q_\alpha$ are defined as

$$\frac{dq_\alpha}{d\lambda} = \Upsilon_\alpha \,. \tag{19}$$

The geodesic relativistic anomaly can be expressed in terms of the Mino angle variable as

$$\psi_r^{\text{geo}}(q_r) = q_r + \Delta\psi_r(q_r) - \Delta\psi_r(0) \,, \tag{20}$$

where

$$\Delta\psi_r(q_r) = \sum_{k \neq 0} \frac{-1}{ik\Upsilon_r} \left( \frac{d\psi_r}{dt} \right)_{q(k)} e^{-ikq_r} \,. \tag{21}$$

The $k$ Fourier mode of a function $\xi(\psi_r)$ is now defined as

$$\xi_{q(k)} = \langle \xi(\psi_r)e^{ikq_r(\psi_r)} \rangle := \frac{1}{2\pi} \int_0^{2\pi} e^{ikq_r(\psi_r)} \xi(\psi_r) \, dq_r \,. \tag{22}$$

## 2.4 Osculating equations

Osculating methods describe accelerating motion of point particles around a given background. They are based on the following observation. At each proper time $\tau$, the true world-line $z^\mu(\tau)$ is tangent to a geodesic $z_G(\tau; I^A(\tau))$ of orbital elements $I^A(\tau) = \{p, e, \psi_r^0, t^0, \phi^0\}$ which evolve with proper time. Here, $\psi_r^0, t^0, \phi^0$ are the initial relativistic anomaly, time and orbital phase, respectively. In the inspiral phase, the evolution timescale is the radiation reaction timescale proportional to $1/\varepsilon$ where $\varepsilon = m_p/M$ is the initial small binary mass ratio. However, it is computationally simpler to switch to the orbital elements $I^A(\tau) = \{p, e, \psi_r, t, \phi\}$ where $\psi_r, t, \phi$ are the evolving relativistic anomaly, time, and orbital phases. The instantaneously tangential geodesics are the osculating orbits. Following [6] the osculating equations read

$$\frac{dp}{dt} = \mathcal{F}_p(p, e, \psi_r),$$ (23a)

$$\frac{de}{dt} = \mathcal{F}_e(p, e, \psi_r),$$ (23b)

$$\frac{d\psi_r}{dt} = f_r(p, e, \psi_r) + \delta f_r(p, e, \psi_r),$$ (23c)

$$\frac{d\phi}{dt} = f_\phi(p, e, \psi_r),$$ (23d)

where $\mathcal{F}_p = O(\varepsilon)$, $\mathcal{F}_e = O(\varepsilon)$ and $\delta f_r = O(\varepsilon)$. The explicit values of the functions are given in Appendix A. In terms of previously defined quantities we have $f_r = f_r^{\text{geo}}(\psi_r)/f_t^{\text{geo}}(r(\psi_r))$ where the dependency in $(p, e)$ was previously understood and now explicit. It is relevant to note that in the approach to the separatrix

$$\lim_{p \to 6+2e} \frac{\mathcal{F}_p}{\mathcal{F}_e} = \frac{2(3+e)}{e-1}.$$ (24)

We denote the self-force components per unit of the secondary mass $m_p$ as

$$f^\mu = \frac{d^2 x^\mu}{d\tau^2} + \Gamma^\mu_{\alpha\beta} \frac{dx^\alpha}{d\tau} \frac{dx^\beta}{d\tau}.$$ (25)

The self-force components appear linearly in Eqs. (23a), (23b) and in the second term on the right-hand side of Eq. (23c). Thanks to the mass-shell constraint $u^2 = -1$, the self-force obeys $f^\alpha u_\alpha = 0$. Thanks to this constraint, complementary to the equatorial plane constraint, we can express the right-hand side of Eq. (23) as the linear combination of two components of the self-force. We choose to combine the components $f^r$ and $f^\phi$ and write

$$\mathcal{F}_p(p, e, \psi_r) = \mathcal{F}_{p,\phi}(p, e, \psi_r) f^\phi(p, e, \psi_r) + \mathcal{F}_{p,r}(p, e, \psi_r) f^r(p, e, \psi_r),$$ (26a)

$$\mathcal{F}_e(p, e, \psi_r) = \mathcal{F}_{e,\phi}(p, e, \psi_r) f^\phi(p, e, \psi_r) + \mathcal{F}_{e,r}(p, e, \psi_r) f^r(p, e, \psi_r),$$ (26b)

$$\delta f_r(p, e, \psi_r) = \delta f_{r,\phi}(p, e, \psi_r) f^\phi(p, e, \psi_r) + \delta f_{r,r}(p, e, \psi_r) f^r(p, e, \psi_r).$$ (26c)

We choose to parameterize the radius, and thus the osculating equations, with the relativistic anomaly $\psi_r$. Instead, we could have used $\varphi_r$ as an angle parameter, which would avoid the use of Eq. (23c). The calculation of the average (15) applied to the osculating equations would also be much simpler. However, using the $\varphi_r$ parametrization must come with knowledge of the geodesic expression $r(\varphi_r)$, which we do not know analytically, see however the recent work [36]. This is the reason why we decided to stick with the parametrization $\psi_r$. The price to pay is the need to use the Jacobian in the average (15).

### 2.5 Asymptotic solution close to the separatrix

Let us introduce the parameter $\delta = p - 6 - 2e$ which can be used to expand around the separatrix $\delta \mapsto 0$ where $e \mapsto e_\star$. We can change variables from time $t$ to $\delta$ in the approach to the separatrix. The relation (24) implies that, in the limit $\delta \mapsto 0$,

$$e(\delta) = e_\star - \frac{1 - e_\star}{8}\delta + o(\delta). \tag{27}$$

It is straightforward to check that the consistent asymptotic solution to the system (23) as $\delta \mapsto 0$ is given by

$$e = e_\star - \frac{1 - e_\star}{8}\sqrt{k_\star}\sqrt{t_\star - t} + o(\sqrt{t_\star - t}), \tag{28}$$

$$p = p_\star + \frac{3 + e_\star}{4}\sqrt{k_\star}\sqrt{t_\star - t} + o(\sqrt{t_\star - t}), \tag{29}$$

$$\psi_r = \psi_{r\star} - \frac{1 + \cos(\psi_{r\star})}{8e_\star \sin(\psi_{r\star})}\sqrt{k_\star}\sqrt{t_\star - t} + o(\sqrt{t_\star - t}), \tag{30}$$

where $p_\star = 6 + 2e_\star$ and $k_\star = -2\lim_{\delta \to 0}(\mathcal{F}_p - 2\mathcal{F}_e)$. Since $\delta'(t) = -\frac{k_\star}{2\delta} + o(\delta^{-1}) < 0$ at leading order in the approach to the separatrix, the self-force quantities are such that $k_\star > 0$. The explicit coefficient is given by

$$k_\star = \frac{4\sqrt{2}(-3 + e_\star)\sqrt{(1 + e_\star)(3 + e_\star)}}{(1 + e_\star \cos(\psi_{r\star}))^2} \tag{31}$$
$$\times \left\{e_\star[-12 + e_\star(-8 + 3e_\star)]\cos(\psi_{r\star}) - 2(2 + e_\star)[-6 + (-4 + e_\star)e_\star + e_\star^2 \cos(2\psi_{r\star})]\right.$$
$$\left. + e_\star^3 \cos(3\psi_{r\star})\right\}\sin^2\left(\frac{\psi_{r\star}}{2}\right)f_\star^\phi + 4\sqrt{2}(-3 + e_\star)\sqrt{-e_\star(1 + e_\star)(-1 + \cos(\psi_{r\star}))}$$
$$\times (-2 - e_\star + e_\star \cos(\psi_{r\star}))\sin(\psi_{r\star})f_\star^r.$$

The radius then reads

$$r = r_\star + o(\sqrt{t_\star - t}), \tag{32}$$

where $r_\star = \frac{p_\star}{1 + e_\star \cos(\psi_{r\star})}$. We find that the intermediate quantities have a typical square root behavior in terms of time but that behavior exactly cancels out at leading order for the evolution of the radius. Since the zero eccentricity case is singular in this parametrization, we cannot relate this leading behavior to the known quasi-circular case [29, 30, 32, 37].

Besides, the subleading analytical behavior of $p(t), e(t)$ and $\psi_r(t)$ can be computed after assuming a half power law in time expansion for all components of the self-force. Substituting these behaviors in (9) reveals the following feature: all terms proportional to the self-force mutually cancel at first subleading order. As a result, despite the presence of self-force effects, the evolution of the radial evolution is given by

$$r = r_\star - \frac{\partial r}{\partial \psi_r}\Big|_\star f_r(p_\star, e_\star, \psi_{r\star})(t_\star - t) + o(t_\star - t) \tag{33}$$

$$= r_\star + \frac{e_\star^{5/2}(e_\star(\cos(\psi_{r\star}) - 1) - 2)}{2(3 + e_\star^2)\sqrt{1 + e_\star}}\sin(\psi_{r\star})|\sin(\psi_{r\star})|(t_\star - t) + o(t_\star - t). \tag{34}$$

This result can be alternatively derived as follows. Even taking into account the self-force, the radial velocity admits the same functional dependency as the geodesic radial velocity, see Eq. (2), while the acceleration depends upon the self-force, see e.g. [29]. At leading order in the approach to the separatrix we therefore have $dr/dt = \pm\sqrt{R}/r^2/\gamma|_\star + o((t - t_\star)^0)$ where

$\gamma|_\star = dt/d\tau|_\star$ is the redshift at the separatrix, see Eq. (79) for its explicit expression. Direct integration over $t$ leads to Eq. (34). The radius increases or decreases upon reaching the separatrix depending upon the sign of $\sin(\psi_{r\star})$, which is coherent with the known phenomenology [22, 25].

## 2.6 Adiabatic equations

We now follow the multiscale expansion framework [2, 3]. We will rederive the adiabatic equations based on the first order differential equations (23).

Let us denote by $p^i$ the set of the two orbital elements $\{p, e\}$. These variables can be treated as independent slow variables in the adiabatic expansion. The angle variables $\varphi_\alpha = \{t, \varphi_r, \varphi_\phi\}$ are the fast variables. We expand the self-force as

$$f^\alpha = \varepsilon f^\alpha_{(1)}(\tilde{t}) + \varepsilon^2 f^\alpha_{(2)}(\tilde{t}) + O(\varepsilon^3), \tag{35}$$

where the slow time is $\tilde{t} := \varepsilon t$. We adopt the following adiabatic expansion

$$\begin{aligned}
\varphi_\alpha(\tilde{t}, \varepsilon) &= \frac{1}{\varepsilon} \left[ \varphi^{(0)}_\alpha(\tilde{t}) + \varepsilon\, \varphi^{(1)}_\alpha(\tilde{t}) + O(\varepsilon^2) \right], \\
p^i(\tilde{t}, \varepsilon) &= p^i_{(0)}(\tilde{t}) + \varepsilon\, p^i_{(1)}(\tilde{t}) + O(\varepsilon^2).
\end{aligned} \tag{36}$$

The adiabatic orbital variables are $p^i_{(0)}(\tilde{t}) = (p_{(0)}(\tilde{t}), e_{(0)}(\tilde{t}))$. The fast variables satisfy

$$\frac{d\varphi_\alpha}{d\tilde{t}} = \frac{1}{\varepsilon} \Omega^{(0)}_\alpha(\tilde{t}) + \varepsilon^0\, \Omega^{(1)}_\alpha(\tilde{t}) + O(\varepsilon). \tag{37}$$

The analysis of the adiabatic expansion [3] leads to the derivation of the 0PA equations. They read

$$\begin{aligned}
\frac{d\varphi^{(0)}_\alpha}{d\tilde{t}} &= \Omega_\alpha(p^j_{(0)}(\tilde{t})), \\
\frac{dp^i_{(0)}}{d\tilde{t}} &= \Gamma_{(1)i}(p^j_{(0)}(\tilde{t})),
\end{aligned} \tag{38}$$

where $\Omega_\alpha(p^j_{(0)})$ are the geodesic formulae of the Boyer-Lindquist fundamental frequencies evaluated as a function of the 0PA expressions of the slow variables. In particular, $\Omega_t = 1$, which is consistent with the fact that $\varphi^{(0)}_t = \tilde{t}$. We have $\Omega^{(0)}(\tilde{t}) = \Omega(p^i_{(0)}(\tilde{t}))$. We also have defined

$$\Gamma_{(1)i}(p^j_{(0)}) := \left\langle \mathcal{F}_p(p^j_{(0)}, \psi_r) \Big|_{f^\alpha \to f^\alpha_{(1)}} \right\rangle, \tag{39}$$

where the average over an orbit is given by Eq. (15). These functions only depend upon the adiabatic slow variables. At 0PA order, the evolution of the slow variables is given by the average of the osculating equations where the self-force term is replaced by its leading $O(\varepsilon)$ order. Moreover, we define the time reversal as $f^\mu(\psi_r) \mapsto \epsilon^\mu f^\mu(-\psi_r)$, $\epsilon^\mu = (-1, 1, 1, -1)$ and the dissipative and conservative parts of the self-force as

$$f^\mu_{\text{diss}} = \frac{1}{2} f^\mu(\psi_r) - \frac{1}{2} \epsilon^\mu f^\mu(-\psi_r), \tag{40}$$

$$f^\mu_{\text{cons}} = \frac{1}{2} f^\mu(\psi_r) + \frac{1}{2} \epsilon^\mu f^\mu(-\psi_r). \tag{41}$$

At 0PA order, only the dissipative part of the self-force contributes, and we have [2, 3]

$$\Gamma_{(1)i}(p^j_{(0)}) := \left\langle \mathcal{F}_p(p^j_{(0)}, \psi_r) \Big|_{f^\alpha \to f^\alpha_{\text{diss}(1)}} \right\rangle. \tag{42}$$

It is interesting to note that the formula (42) admits a nice quasicircular limit when we set $e_{(0)} = 0$ before computing the average. In fact, in this limit, we obtain $\varphi_\phi = \phi$, $\frac{de_{(0)}}{d\tilde{t}} = 0$ and

$$\frac{dp_{(0)}}{d\tilde{t}} = \frac{2(p_{(0)} - 3)^2 p_{(0)}^{3/2}}{p_{(0)} - 6} f^\phi(p_{(0)}).$$
(43)

This formula exactly matches with the Schwarzschild case [30, 31, 38]. In the quasi-circular limit, $p_{(0)}$ is interpreted as the reduced radius of the shrinking orbital circle. When the secondary approaches the ISCO, the time-domain evolution of its radius is asymptotically given by $p_{(0)} = 6 + 6^{7/4} \sqrt{-f_*^\phi (t_\star - t)}$ for $t < t_\star$, where $t_\star$ is the time at which the ISCO is crossed and $f_*^\phi < 0$ is the $\phi$ component of the self-force evaluated at the ISCO.

## 2.7 Adiabatic equations: Expansion in Fourier modes

In this section, we compute the driving force $\Gamma_{(1)i}$ of the orbital elements at adiabatic order (42). We assume that the self-force is periodic in $\psi_r$ and that it admits a Fourier series expansion such that we can write

$$f^\mu(p^i, \psi_r) = \sum_{k \in \mathbb{Z}} f_{(k)}^\mu(p^i) e^{-ik\psi_r},$$
(44)

where $f_{(k)}^\mu = \frac{1}{2\pi} \int_0^{2\pi} f^\mu(p^i, \psi_r) e^{ik\psi_r} d\psi_r$. Inserting this expansion into Eq. (26), the right-hand sides of the osculating equations for the orbital elements $p^i$ now contain a linear sum of self-force Fourier modes. The average, and thus the 0PA equations, can be computed. These equations are

$$\frac{dp_{(0)}}{d\tilde{t}} = \sum_{k \in \mathbb{Z}} \left( \Gamma_{(1)p(k)}^\phi(p_{(0)}^j) + \Gamma_{(1)p(k)}^r(p_{(0)}^j) \right),$$
$$\frac{de_{(0)}}{d\tilde{t}} = \sum_{k \in \mathbb{Z}} \left( \Gamma_{(1)e(k)}^\phi(p_{(0)}^j) + \Gamma_{(1)e(k)}^r(p_{(0)}^j) \right),$$
(45)

where

$$\Gamma_{(1)p(k)}^\phi = \left\langle \mathcal{F}_{p,\phi}(p_{(0)}^j, \psi_r) e^{-ik\psi_r} f_{(1)(k)}^\phi(p_{(0)}^j) \right\rangle,$$
$$\Gamma_{(1)p(k)}^r = \left\langle \mathcal{F}_{p,r}(p_{(0)}^j, \psi_r) e^{-ik\psi_r} f_{(1)(k)}^r(p_{(0)}^j) \right\rangle,$$
$$\Gamma_{(1)e(k)}^\phi = \left\langle \mathcal{F}_{e,\phi}(p_{(0)}^j, \psi_r) e^{-ik\psi_r} f_{(1)(k)}^\phi(p_{(0)}^j) \right\rangle,$$
$$\Gamma_{(1)e(k)}^r = \left\langle \mathcal{F}_{e,r}(p_{(0)}^j, \psi_r) e^{-ik\psi_r} f_{(1)(k)}^r(p_{(0)}^j) \right\rangle.$$
(46)

It turns out that these averages can be analytically computed for any value of the integer $k$. We shall not display the expressions $\Gamma_{(1)p(k)}^\phi$ and $\Gamma_{(1)e(k)}^\phi$ because they are lengthy. However, the interested reader can consult the notebook associated to this paper on Github where the adiabatic equations are explicitly written. It is however easier to compute $\Gamma_{(1)p(k)}^r$ and $\Gamma_{(1)e(k)}^r$. Their analytical expressions are

$$\Gamma_{(1)p(k)}^r = -\frac{\Omega_r^{(0)}}{2\pi} \frac{2 e_{(0)} p_{(0)}^3 (3 + e_{(0)}^2 - p_{(0)})}{4e_{(0)}^2 - (p_{(0)} - 6)^2} \mathcal{K}(e_{(0)}; k) f_{(1)(k)}^r(p_{(0)}, e_{(0)}),$$

$$\Gamma_{(1)e(k)}^r = -\frac{\Omega_r^{(0)}}{2\pi} \frac{(3 + e_{(0)}^2 - p_{(0)})(6 + 2e_{(0)}^2 - p_{(0)}) p_{(0)}^2}{4e_{(0)}^2 - (p_{(0)} - 6)^2} \mathcal{K}(e_{(0)}; k) f_{(1)(k)}^r(p_{(0)}, e_{(0)}),$$
(47)

where $\mathcal{K}(e_{(0)}; k) := \int_0^{2\pi} \frac{\sin(\psi)}{(1+e_{(0)}\cos(\psi))^2} e^{-ik\psi} d\psi = -i \int_0^{2\pi} \frac{\sin(\psi)\sin(k\psi)}{(1+e_{(0)}\cos(\psi))^2} d\psi$ can be computed analytically and does not diverge at the separatrix for any value of $k$. To do so, we first write $\sin(k\psi)$ as given by the formula [39]

$$\sin(k\,\psi) = k\,\cos^{k-1}(\psi)\,\sin(\psi) - \binom{k}{3}\cos^{k-3}(\psi)\,\sin^3(\psi) + \binom{k}{5}\cos^{k-5}(\psi)\,\sin^5(\psi) - \dots \quad (48)$$

Then, thanks to the change of variables $z := \tan(\frac{\psi}{2})$ and trigonometric reductions, it is possible to write the integrand as the fraction of polynomials of $z$ which can be explicitly integrated. In particular, $\mathcal{K}(e_{(0)}; 0) = 0$. Therefore, the average radial self-force does not contribute to the driving force at adiabatic order.

The post-adiabatic equations can be derived following the Hinderer-Flanagan analysis [3] using the fast variable $\psi_r$. However, since $f_r$ depends upon $\psi_r$, the post-adiabatic equations are coupled between Fourier modes $k$ and are not therefore easily solvable. We shall not develop them here.

## 3 Analytic expansion towards the separatrix

In this section, we asymptotically develop the expansion of the equations (45) in the approach towards the separatrix. For that purpose, we define the parameter $\delta_{(0)}$ at adiabatic order, which identically vanishes at the separatrix, as

$$\delta_{(0)} \equiv p_{(0)} - (6 + 2e_{(0)}). \quad (49)$$

where $p_{(0)}$ and $e_{(0)}$ are defined in Eq. (36). We will study the limit $\delta_{(0)} \to 0^+$. In that limit, we expect the eccentricity to asymptote to a finite value $e_{(0)} \to e_\star$, which is interpreted as the eccentricity at the separatrix crossing. We have immediately

$$\frac{d\delta_{(0)}}{d\tilde{t}} = \sum_{k \in \mathbb{Z}} \left( \Gamma^\phi_{(1)\delta(k)}(\delta_{(0)}, e_{(0)}) + \Gamma^r_{(1)\delta(k)}(\delta_{(0)}, e_{(0)}) \right), \quad (50)$$

where

$$\begin{aligned}
\Gamma^\phi_{(1)\delta(k)} &= \Gamma^\phi_{(1)p(k)} - 2\Gamma^\phi_{(1)e(k)}, \\
\Gamma^r_{(1)\delta(k)} &= \Gamma^r_{(1)p(k)} - 2\Gamma^r_{(1)e(k)},
\end{aligned} \quad (51)$$

have been rewritten in terms of $\delta_{(0)}$ and $e_{(0)}$.

We now assume that the self-force $f^\mu_{(1)}$ as a function of the energy $E$ and angular momentum $L$ is continuous and differentiable at the separatrix. The relationship between the elements $(E, L)$ and $(e, p)$ (understood as $(e_{(0)}, p_{(0)})$) is given in Eq. (6). The Jacobian is singular at the separatrix. Given the ratio $dp/de$ at the separatrix (24), we have

$$\partial_{\delta_{(0)}} f^\mu_{(1)}(\delta_{(0)}, e_{(0)})|_{\delta_{(0)}=0^+} = \frac{e_{(0)}-1}{8} \partial_{e_{(0)}} f^\mu_{(1)}(\delta_{(0)}, e_{(0)})|_{\delta_{(0)}=0^+}. \quad (52)$$

### 3.1 Expansion of elliptic integrals

We now aim to find the leading order solution in the $\delta_{(0)} \to 0$ expansion. The major difficulty that we encounter is the following. The orbit average quantities $\Gamma^\phi_{(1)\delta(k)}$ and $\Gamma^\phi_{(1)e(k)}$

depend upon complete elliptic integrals of the third kind such as $\Pi\left(\frac{2e(2+2e+\delta)}{(1+e)(4e+\delta)}, \frac{4e}{4e+\delta}\right)$ and $\Pi\left(\frac{16e}{(4+\delta)(4e+\delta)}, \frac{4e}{4e+\delta}\right)$ (here we substituted $e_{(0)}$ by $e$ and $\delta_{(0)}$ by $\delta$ in order to simplify the notation). The asymptotic development of these functions as $\delta \to 0$ is not documented in standard textbooks of special functions. We are however able to prove that, at leading order in $\delta$, and as long as $1 \geq e > 0$,

$$
\Pi\left(\frac{2e(2+2e+\delta)}{(1+e)(4e+\delta)}, \frac{4e}{4e+\delta}\right) = \frac{1}{\delta} \frac{2\sqrt{2\,e(e+1)}}{\sqrt{(3+2e)(1+4e)}} \tag{53}
$$
$$
\times \left( \Pi\left(\frac{2e(3+2e)}{(1+e)(1+4e)}, \frac{4e}{1+4e}\right) + \sqrt{(3+2e)(1+4e)}\,\Lambda(e) \right)
$$
$$
+ \frac{1+\log(64e)-\log(\delta)}{4} + \frac{1+3e}{2\sqrt{2}\,\sqrt{e(1+e)(3+2e)(1+4e)}}
$$
$$
\times \left[ \Pi\left(\frac{2e(3+2e)}{(1+e)(1+4e)}, \frac{4e}{1+4e}\right) + \sqrt{(3+2e)(1+4e)}\,\Lambda(e) \right]
$$
$$
+ O(\delta),
$$
$$
\Pi\left(\frac{16e}{(4+\delta)(4e+\delta)}, \frac{4e}{4e+\delta}\right) = \frac{1}{\delta} \frac{2\left(\sqrt{5e(1+4e)(5+4e)}\,\Pi\left(\frac{16e}{5+20e}, \frac{4e}{1+4e}\right) + 5\sqrt{e}(1+4e)\Theta(e)\right)}{5\sqrt{1+e}(1+4e)}
$$
$$
+ \frac{1-e}{4(1+e)} + \frac{\log(64e)-\log(\delta)}{4} + \frac{(1+e+e^2)\Theta(e)}{4\sqrt{e}(1+e)^{3/2}}
$$
$$
+ \frac{\Pi\left(\frac{16e}{5+20e}, \frac{4e}{1+4e}\right)\sqrt{(1+e)(1+4e)(5+4e)}}{4\sqrt{5}\,e(1+e)^2(1+4e)(5+4e)}
$$
$$
\times \left( 5e^{1/2} + 9e^{3/2} + 9e^{5/2} + 4e^{7/2} \right) + O(\delta),
$$

where we have defined the functions

$$
\Lambda(e) \equiv -\int_0^1 \frac{(1+e)K\left(\frac{4e}{4e+x}\right)}{(2+2e+x)^{3/2}\sqrt{4e+x}}dx\,,
$$
$$
\Theta(e) \equiv \int_0^1 \frac{(4e+x)E\left(\frac{4e}{4e+x}\right) - 2(2+2e+x)K\left(\frac{4e}{4e+x}\right)}{2\sqrt{(4e+x)(4+x)(4+4e+x)}}dx\,. \tag{54}
$$

These are pure functions of the eccentricity which are well defined throughout the entire range $0 \leq e \leq 1$ and admit finite $e = 0$ and $e = 1$ values. The zero eccentricity values are $\Lambda(0) = -\frac{\pi}{2\sqrt{3}}$ and $\Theta(0) = -\frac{\pi}{2}$. The $e = 1$ values are

$$
\Lambda(1) = \frac{1}{25}\pi\,_2F_1\left(\frac{3}{2}, \frac{3}{2}; 2; \frac{4}{5}\right) - 1 = -1 + E\left(\frac{4}{5}\right) - \frac{1}{5}K\left(\frac{4}{5}\right) \approx 0.2730\,,
$$
$$
\Theta(1) = \int_0^1 \frac{E\left(\frac{4}{4+x}\right) - 2K\left(\frac{4}{4+x}\right)}{2\sqrt{8+x}}dx \approx -0.7315\,, \tag{55}
$$

where $_2F_1(a, b, c, z)$ is the Gaussian hypergeometric function (see [40] for elements on hypergeometric functions). We note that the two expansions truncated at subleading order (53) become better fits of the exact functions at high values of the eccentricity. We detail in Appendix B the derivation of these series expansion as well as further relevant identities and plots.

Once the elliptic integrals are replaced by Eq. (53), we can proceed to the $\delta_{(0)}$-expansion of the adiabatic equations and derive their behaviour as the inspiral approaches the separatrix. This is the content of the next section.

## 3.2 Asymptotic adiabatic equations

Let us now present the resulting asymptotic expansion of the adiabatic equations in the limit $\delta_{(0)} \to 0^+$. We could use $\delta_{(0)}$ as a substitute for the slow time $\tilde{t}$. In other words, we could define the functions $\tilde{t} = \tilde{t}(\delta_{(0)})$ and $e_{(0)}(\delta_{(0)}) = e_{(0)}(\tilde{t}(\delta_{(0)}))$. Given Eq. (24), we have at the separatrix

$$\frac{\frac{d e_{(0)}(\tilde{t})}{d\tilde{t}}}{\frac{d\delta_{(0)}(\tilde{t})}{d\tilde{t}}}\Bigg|_{\delta_{(0)}=0} = \frac{d e_{(0)}(\delta_{(0)})}{d\delta_{(0)}}\Bigg|_{\delta_{(0)}=0} = \frac{e_\star - 1}{8}\,, \tag{56}$$

where $e_\star$ is the value of the eccentricity at the separatrix. This tells us that the eccentricity $e_{(0)}$ viewed as a function of $\delta_{(0)}$ has $e_{(0)}(0) = e_\star$ and $e'_{(0)}(0) = \frac{e_\star - 1}{8}$. With this at hand, we are ready to derive the expansions of the 0PA equations.

**Fast angle variables**   We first expand the evolution of the angle variables given in Eq. (38). This amounts to compute the asymptotic behavior of the Boyer-Lindquist geodesic frequencies $\Omega_r^{(0)}$ and $\Omega_\phi^{(0)}$. At leading order, we find

$$\begin{aligned}
\Omega_r^{(0)} &= \frac{d\varphi_r^{(0)}}{d\tilde{t}} = -\frac{(e_\star(1+e_\star))^{3/2}\pi}{2(3+e_\star)^2}\frac{1}{e_\star \log(\delta_{(0)}) + \mathcal{B}(e_\star)} + o(\delta_{(0)}^0)\,,\\
\Omega_\phi^{(0)} &= \frac{d\varphi_\phi^{(0)}}{d\tilde{t}} = \frac{e_\star}{2\sqrt{2}}\left(\frac{1+e_\star}{3+e_\star}\right)^{3/2}\frac{\log(\delta_{(0)}) - \log(64 e_\star)}{e_\star \log(\delta_{(0)}) + \mathcal{B}(e_\star)} + o(\delta_{(0)}^0)\,.
\end{aligned} \tag{57}$$

The function $\mathcal{B}(e)$ is real in the range $0 \le e < 1$ and admits two roots at $e = 0$ and $e \approx 0.01433$. It is positive in the range $0 < e < 0.01433$ and negative otherwise. In the low eccentricity limit we have $\mathcal{B}(e) = -e\log(64e) + O(e^2)$. In the high eccentricity limit, we have that $\mathcal{B}(e) = -\frac{\pi}{\sqrt{2}}(1-e)^{-3/2} + O((1-e)^{-1/2})$. The complete expression and plots of $\mathcal{B}$ can be found in Appendix C. Given the denominator $e_\star \log(\delta_{(0)}) + \mathcal{B}(e_\star)$, the domain of validity of this expansion is $0 < \delta_{(0)} < e^{-\mathcal{B}(e_\star)/e_\star}$, which reduces to $0 < \delta_{(0)} < 64 e_\star$ in the small eccentricity limit. The expansion therefore is not valid in the strict zero eccentricity limit.[1] The next-to-leading order of (57) is computed and can be found in the Mathematica notebook of this paper.

At finite $e_\star$, the limit $\delta_{(0)} \to 0^+$ of the equations gives

$$\Omega_{r_\star}^{(0)} = 0\,, \qquad \Omega_{\phi\star}^{(0)} = \frac{1}{2\sqrt{2}}\left(\frac{1+e_\star}{3+e_\star}\right)^{3/2}\,. \tag{58}$$

These latter equations have a zero eccentricity limit where $\Omega_{\phi\star}^{(0)} = \frac{1}{6\sqrt{6}}$, as the quasi-circular case.

Given the known asymptotic behaviour of the fundamental frequencies $\Omega_r^{(0)}$ and $\Omega_\phi^{(0)}$ we can derive the instantaneous number of whirls the secondary body achieves during the late inspiral. These cycles, first discussed in [41], are mostly occurring as a whirl motion around the peripasis, where the secondary body is the closest to the primary black hole. This instantaneous number of whirling cycles is given by

$$N_w := \frac{\phi(\psi_r = 2\pi) - \phi(\psi_r = 0)}{2\pi} = \frac{\Omega_\phi^{(0)}}{\Omega_r^{(0)}} = \sqrt{\frac{3+e_\star}{e_\star}}\frac{\log(64 e_\star) - \log(\delta_{(0)})}{\sqrt{2}\,\pi} + o(\delta_{(0)}^0)\,. \tag{59}$$

This expression explicitly shows that the instantaneous number of whirls increases in a logarithmic manner as we come closer and closer to the end of the inspiral. This confirms the computation produced in [41]. The graphic representation of $N_w$ is given in Fig. 1.

---

[1]The small eccentricity limit could be studied instead by performing a post-adiabatic expansion of the low eccentricity expansion of the osculating equations formulated in the appropriate $(\alpha, \beta)$ formulation [6]. Such equations, which could be compared with the quasi-circular case, are not studied here.

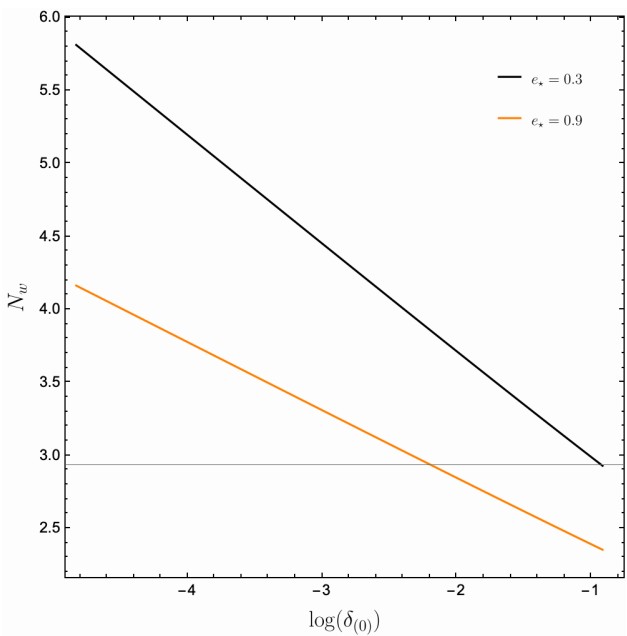

Figure 1: Plot of $N_w$ as a function of $\log(\delta_{(0)})$. The black line corresponds to $e_\star = 0.3$, while the red line corresponds to $e_\star = 0.9$. These results can be directly compared with [41], where a similar plot is presented, but on linear scale on the horizontal axes. As expected, the instantaneous number of whirls, $N_w$, increases sharply as the system approaches the separatrix, diverging to infinity at $\log(\delta_{(0)}) \to -\infty$. Specifically, for $\delta_{(0)} = 0.01$ ($\log(\delta_{(0)}) = -4.61$), we find $N_w(0.01) = 5.645$ for $e = 0.3$ and $N_w(0.01) = 4.058$ for $e = 0.9$.

**Slow orbital variables**    The equations that drive the evolution of $\delta_{(0)}$ and $e_{(0)}$ read

$$\frac{d\delta_{(0)}}{d\tilde{t}} = \frac{\mathcal{A}(e_\star)}{\delta_{(0)}\big(e_\star \log(\delta_{(0)}) + \mathcal{B}(e_\star)\big)} + O(\delta_{(0)}^0), \tag{60a}$$

$$\frac{de_{(0)}}{d\tilde{t}} = \frac{e_\star - 1}{8} \frac{\mathcal{A}(e_\star)}{\delta_{(0)}\big(e_\star \log(\delta_{(0)}) + \mathcal{B}(e_\star)\big)} + O(\delta_{(0)}^0). \tag{60b}$$

The constant $\mathcal{A}$ can be decomposed as $\mathcal{A} = \sum_{k\in\mathbb{Z}}\big(\mathcal{A}_{r(k)}(e_\star)f^r_{(1)(k)\star} + \mathcal{A}_{\phi(k)}(e_\star)f^\phi_{(1)(k)\star}\big)$. We checked explicitly this structure for the Fourier modes ranging from $k = -2$ to $k = 2$ and we expect that it holds for any $k \in \mathbb{Z}$. It turns out that $\mathcal{A}_{r(0)} = 0$, $\mathcal{A}_{r(k)} = -\mathcal{A}_{r(-k)}$ and $\mathcal{A}_{\phi(k)} = \mathcal{A}_{\phi(-k)}$. The explicit expressions of $\mathcal{A}_{r(k)}$, $\mathcal{A}_{\phi(0)}$, $\mathcal{A}_{\phi(\pm 1)}$ and $\mathcal{A}_{\phi(\pm 2)}$ are given in Appendix C. In these equations, the $\mathcal{B}(e_\star)$ factor arises from the asymptotic behavior of $\Omega_r$. The $\mathcal{A}$ factor arises from the asymptotic expansion of $\mathcal{F}_p - 2\mathcal{F}_e$ and $\mathcal{F}_e$. The subleading orders of Eq. (60a) and (60b) are explicitly given in Appendix D. The relations (60a) and (60b) approximate well the 0PA equations for small $\delta_{(0)}$; the plots in Fig. 2 compare the approximations with the analytical formulae for $\delta_{(0)} < 0.4$.

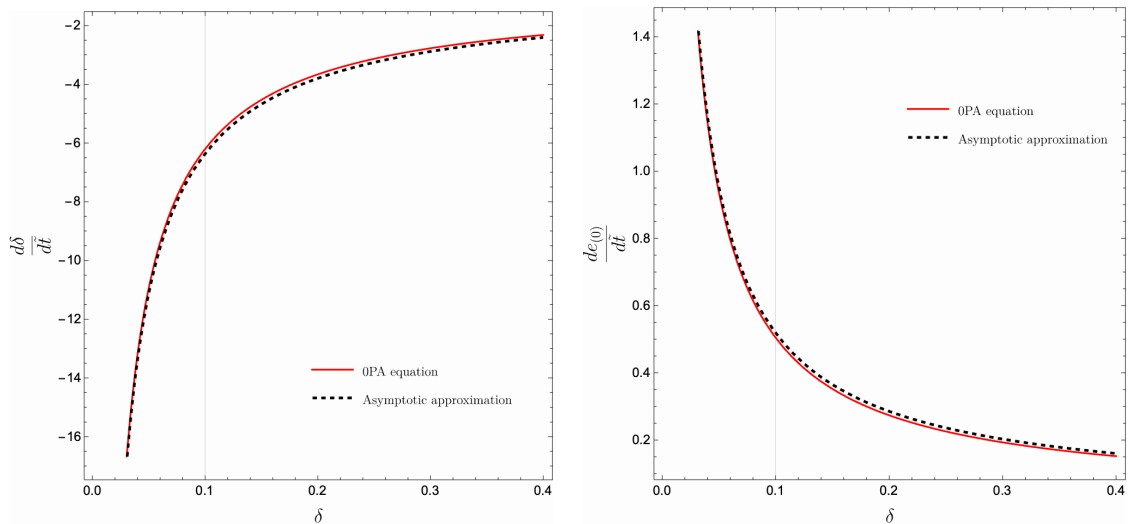

Figure 2: Comparison of the complete 0PA equations (solid lines) with their asymptotic expansions (dashed lines). The quantities $\frac{d\delta_{(0)}}{d\tilde{t}}\Big|_{\text{0PA}}$ (left panel) and $\frac{de_{(0)}}{d\tilde{t}}\Big|_{\text{0PA}}$ (right panel) show a strong agreement with their respective asymptotic expansions for $\delta_{(0)} < 0.4$. In generating these plots, the eccentricity $e_{(0)}$ was set to 0.2673, with self-force data at the separatrix for this value of $e_{(0)}$ kindly provided by Maarten van de Meent. For $\delta_{(0)} = 0.4$, the relative mismatches between the numerical and asymptotic values are approximately 3% for $\frac{de_{(0)}}{d\tilde{t}}$ and 0.3% for $\frac{d\delta_{(0)}}{d\tilde{t}}$.

**Rewriting of the fast angle variables**  The differential equations for the fast variables (57) can be written as $\frac{d\varphi_\alpha^{(0)}}{d\delta_{(0)}}$ as well. Thanks to (60a), we have

$$
\begin{aligned}
\frac{d\varphi_r^{(0)}}{d\delta_{(0)}} &= -\frac{(e_\star(1+e_\star))^{3/2}\pi}{2(3+e_\star)^2\,\mathcal{A}(e_\star)}\delta_{(0)} + o(\delta_{(0)}), \\
\frac{d\varphi_\phi^{(0)}}{d\delta_{(0)}} &= -\frac{1}{2\sqrt{2}}\left(\frac{1+e_\star}{3+e_\star}\right)^{3/2}\frac{e_\star\big(\log(64)+\log(e_\star)-\log(\delta_{(0)})\big)\delta_{(0)}}{\mathcal{A}(e_\star)} + o(\delta_{(0)}).
\end{aligned}
\tag{61}
$$

We can also develop an adiabatic equation involving $\delta_{(0)}$ and the anomalous phase $\psi_r$ while considering $\psi_r(\delta_{(0)})$. For that purpose, we consider Eq. (60a) and the 0PA term of Eq. (23c). This gives

$$
\frac{d\psi_r}{d\delta_{(0)}} = -\delta_{(0)}\frac{\sqrt{e_\star}(1+e_\star\cos(\psi_r))^2(-2-e_\star+e_\star\cos(\psi_r))(e_\star\log(\delta_{(0)})+\mathcal{B}(e_\star))\sin(\frac{\psi_r}{2})}{4\varepsilon\sqrt{1+e_\star}(3+e_\star)^2\mathcal{A}(e_\star)} + o\big(\delta_{(0)}\big).
\tag{62}
$$

## 3.3 Asymptotic analytical solutions

It is possible to solve the asymptotic equations analytically thanks, notably, to the real branch $k = -1$ of the Lambert $W$ function $W_{-1}(\cdot)$.

First of all, we focus on the solution of Eq. (60a) for $\delta_{(0)}(\tilde{t})$. The solution is

$$
\delta_{(0)}(\tilde{t}) = e^{\frac{1}{2}-\frac{\mathcal{B}(e_\star)}{e_\star}}e^{\frac{1}{2}W_{-1}\left(-\frac{4\mathcal{A}}{e_\star}e^{\frac{2\mathcal{B}(e_\star)}{e_\star}-1}(\tilde{t}_\star-\tilde{t})\right)},
\tag{63}
$$

and is valid for $\tilde{t} < \tilde{t}_\star$, where $\tilde{t}_\star$ is the separatrix crossing time. In this solution, $W_{-1}(z)$ is the $-1$ type real branch of the $W$ Lambert function defined as being the reciprocal of $z = W(z)e^{W(z)}$.

Elements on this type of function can be found in the Appendix E. The function $\delta_{(0)}(\tilde{t})$ is monotonously decreasing with slow time and reaches 0 at $\tilde{t} = \tilde{t}_\star$. We can invert Eq. (63) and obtain

$$
\begin{aligned}
\tilde{t}(\delta_{(0)}) = \tilde{t}_\star + \delta_{(0)}^2 \frac{e_\star}{4\mathcal{A}(e_\star)} &\left[ \left( \frac{2\mathcal{B}(e_\star)}{e_\star} - 1 \right) + \log(\delta_{(0)}) \right] + \delta_{(0)}^3 \\
\times &\left[ -\frac{8\big(9\mathcal{E}_\delta(e_\star) - 3\mathcal{F}_\delta(e_\star) + 2\mathcal{G}_\delta(e_\star)\big) - 3(e_\star - 1)\big((e_\star - 3\mathcal{B}(e_\star))\mathcal{A}'(e_\star) + \mathcal{A}(e_\star)(-1 + 3\mathcal{B}'(e_\star))\big)}{216\mathcal{A}^2(e_\star)} \right. \\
&\left. + \frac{-24\mathcal{F}_\delta(e_\star) + 16\mathcal{G}_\delta(e_\star) + 3(e_\star - 1)\big(\mathcal{A}(e_\star) - e_\star\mathcal{A}'(e_\star)\big)}{72\mathcal{A}^2(e_\star)} \log(\delta_{(0)}) - \frac{\mathcal{G}_\delta(e_\star)}{3\mathcal{A}^2(e_\star)} \log^2(\delta_{(0)}) \right] \\
+ O(\delta_{(0)}^4 &\log^3 \delta_{(0)}) .
\end{aligned}
\tag{64}
$$

Here we used the subleading orders of the equations derived in Appendix D to compute the residual of the leading order solution.

The solution of Eq. (D.2) can be found. In terms of $\delta_{(0)}$ the eccentricity is given by

$$
e_{(0)}(\delta_{(0)}) = e_\star + \frac{e_\star - 1}{8}\delta_{(0)} + \frac{8\mathcal{G}_e(e_\star) + (1 - e_\star)\mathcal{G}_\delta(e_\star)}{16\, e_\star \mathcal{A}(e_\star)} \delta_{(0)}^2 \log(\delta_{(0)}) + O(\delta_{(0)}^2) .
\tag{65}
$$

This relation tells us that the eccentricity increases as the secondary black hole approaches the separatrix. Such behavior in the strong-field regime is consistent with expectations, as it is described in [14, 17, 42].

After analysis, we conjecture that the coefficient appearing in (65) can be written in terms of the self-force evaluated at the periapsis,

$$
8\mathcal{G}_e(e_\star) + (1 - e_\star)\mathcal{G}_\delta(e_\star) = \frac{2\sqrt{2}\, e_\star (3 + e_\star)^{3/2} (3 - e_\star)^2}{\sqrt{1 + e_\star}} f_{(1)}^\phi(p_\star, e_\star, 0) .
\tag{66}
$$

We proved that this identity holds when including all Fourier modes $k = -2, -1, 0, 1, 2$ and we conjecture that it holds more generally.

Finally, the solutions of the leading order differential equations for the angle variable (61) read as

$$
\begin{aligned}
\varphi_r^{(0)}(\delta_{(0)}) = \varphi_{r\star} &- \frac{(e_\star(e_\star + 1))^{3/2}\pi}{4(3 + e_\star)^2 \mathcal{A}(e_\star)} \delta_{(0)}^2 \\
&+ \frac{\sqrt{1 + e_\star}\,\big(3\,e_\star^{3/2} + 4e_\star^{5/2} + e_\star^{7/2}\big)\pi\mathcal{G}_\delta(e_\star)}{6e_\star(3 + e_\star)^3 \mathcal{A}^2(e_\star)} \delta_{(0)}^3 \log(\delta_{(0)}) + O(\delta_{(0)}^3) , \\
\varphi_\phi^{(0)}(\delta_{(0)}) = \varphi_{\phi\star} &- \frac{1}{2\sqrt{2}} \left( \frac{1 + e_\star}{3 + e_\star} \right)^{3/2} \frac{e_\star \big(1 + \log\big(4096\,e_\star^2\big) - 2\log(\delta_{(0)})\big)}{4\,\mathcal{A}(e_\star)} \delta_{(0)}^2 \\
&- \left( \frac{1 + e_\star}{3 + e_\star} \right)^{3/2} \frac{\mathcal{G}_\delta(e_\star)}{6\sqrt{2}\mathcal{A}^2(e_\star)} \delta_{(0)}^3 \log^2(\delta_{(0)}) + O(\delta_{(0)}^3 \log(\delta_{(0)})) ,
\end{aligned}
\tag{67}
$$

where $\varphi_{\phi\star}$ and $\varphi_{r\star}$ are the angle variables at the separatrix.

Furthermore, as $\delta_{(0)}(\tilde{t})$ and $e_{(0)}(\delta_{(0)}(\tilde{t}))$ have been solved, we can obtain the 0PA relativistic anomaly $\psi_{r(0)}$. This latter is driven by the 0PA part of Eq. (23c), that is the $O(\varepsilon^0)$ term of this equation where $e, p$ and $\psi_r = \varepsilon^{-1}\psi_{r(0)} + O(\varepsilon^0)$ have been restricted to their adiabatic expressions

$$
\frac{d\psi_{r(0)}}{d\tilde{t}} = \mathsf{f}_r(6 + 2e_{(0)} + \delta_{(0)}, e_{(0)}, \psi_{r(0)}) .
\tag{68}
$$

This formula is consistent with the statement that the mapping $\psi_r(\varphi_r)$ established at the geodesic level, see the paper [36], still holds at adiabatic order but with the replacement

$e \to e_{(0)}, p \to p_{(0)}$ and $\psi_r \to \psi_{r(0)}$. As a consequence, $\psi_{r(0)}(\tilde{t})$ deviates from its geodesic behavior due to the slow time dependency of $e_{(0)}$ and $p_{(0)}$. Let us now substitute $\delta_{(0)}(\tilde{t})$ and $e_{(0)}(\delta_{(0)}(\tilde{t}))$ into Eq. (68). No pole arises in this expansion. The leading term reduces to Eq. (68) with $e_{(0)}$ replaced by $e_\star$ and $\delta_{(0)}$ set to 0. Hence, we can directly integrate and obtain

$$\psi_{r(0)} = \psi_{r\star} + f_r(6 + 2e_\star, e_\star, \psi_{r\star})(t - t_\star) + O(t - t_\star)^2 + O(\varepsilon), \tag{69}$$

where $\psi_{r\star}$ is the value of $\psi_r$ at the separatrix. This result is the first order Taylor development of the anomalous phase around its value at the separatrix in the adiabatic approximation. We can also express $\psi_{r(0)}$ as a function of $\delta_{(0)}$ after using Eq. (64).

Let us now briefly comment upon the 1PA order. At 1PA order, the term $\delta f_r(p, e, \psi_r)$ in Eq. (23c) becomes relevant. Substituting the 0PA solution, we find a pole as $\delta_{(0)} \mapsto 0$ and the 1PA ($O(\varepsilon)$) solution contains therefore a term more leading than $(t - t_\star)$ in the expansion (69). We conclude that higher PA terms become more divergent than the 0PA terms in the limit to the separatrix, as in the quasi-circular case [30].

## 3.4 Consequences

Up to this point, we have derived the adiabatic equations of motion near the last stable orbit and obtained their corresponding solutions. This information improves our understanding of the dynamics. For example, it allows us to determine how the energy and angular momentum evolve before crossing the separatrix, but also how the periapsis and the apoapsis behave asymptotically. Ultimately, we have all the tools required to compute the late evolution of the radius and the dynamics of the late inspiral in the $(E, L)$ and $(p, e)$ phase spaces.

**Energy and momentum**   The (adimensionalized) energy and angular momentum are functions of the orbital elements $e$ and $\delta$ as given in Eq. (6). In the approach to the separatrix, $E(\delta_{(0)})$ and $L(\delta_{(0)})$ behave as

$$E(\delta_{(0)}) = E_\star(e_\star) - \frac{8\mathcal{G}_e(e_\star) + (1 - e_\star)\mathcal{G}_\delta(e_\star)}{4\sqrt{2}(-3 + e_\star)(3 + e_\star)\sqrt{9 - e_\star^2}\,\mathcal{A}(e_\star)} \delta_{(0)}^2 \log(\delta_{(0)}) + O(\delta_{(0)}^2), \tag{70}$$

$$L(\delta_{(0)}) = L_\star(e_\star) + \frac{8\mathcal{G}_e(e_\star) + (1 - e_\star)\mathcal{G}_\delta(e_\star)}{2(3 + 2e_\star - e_\star^2)^{3/2}\mathcal{A}(e_\star)} \delta_{(0)}^2 \log(\delta_{(0)}) + O(\delta_{(0)}^2), \tag{71}$$

where the $E_\star$ and $L_\star$ (their value at the separatrix) are given by Eq. (7) and (8) with $e$ replaced by $e_\star$. The quantities $\mathcal{G}_e(e_\star)$ and $\mathcal{G}_\delta(e_\star)$ appear at subleading order in the expansion close to the separatrix. They are defined in Appendix D. We deduce in particular that

$$\frac{dE}{dL} = \Omega_{\phi\star} + \frac{(-3 + e_\star)e_\star\sqrt{1 + e_\star}\,\mathcal{A}(e_\star)}{16\sqrt{2}\,\sqrt{3 + e_\star}\,\big(8\mathcal{G}_e(e_\star) + (1 - e_\star)\mathcal{G}_\delta(e_\star)\big)\log(\delta_{(0)})} + O(\delta_{(0)}). \tag{72}$$

Recall that $\Omega_{\phi\star}$ is given by the second equality in Eq. (58). This relation can be seen as a generalization of what is known for the quasicircular case; in the late inspiral, the rates of change of $E$ and $L$ are directly given by the value of the azimuthal fundamental frequency at the separatrix. In particular, the limit $e_\star \to 0$ provides $\frac{dE}{d\delta_{(0)}} = \frac{1}{6\sqrt{6}}\frac{dL}{d\delta_{(0)}} + O(\delta_{(0)})$.

**Critical function**   As defined in [42], the so-called *critical function* is a function that compares the evolution of the semi-latus rectum to that of the eccentricity. It is defined by

$$C(p, e) := \frac{d\log(e)}{d\log(p)}. \tag{73}$$

The computation of this function is immediate. Near the separatrix crosssing, it is a function of $\delta_{(0)}$ only. We obtain

$$C(\delta_{(0)}) = -\frac{1-e_\star}{e_\star} + \frac{4\big(8\mathcal{G}_e(e_\star)+(1-e_\star)\mathcal{G}_\delta(e_\star)\big)}{e_\star^2(3+e_\star)\mathcal{A}(e_\star)}\delta_{(0)}\log(\delta_{(0)}) + O(\delta_{(0)}), \qquad (74)$$

where the higher order in $\delta_{(0)}$ is available in the notebook. This result reproduces what has been computed in [42]. In this paper, this computation was used to highlight the fact that, as $0 < e_\star < 1$, the critical function remains negative regardless of the value of the eccentricity at the separatrix. This implies that, as long as the semi-latus rectum decreases, the eccentricity increases at the late inspiral near the separatrix. This is a characteristic behavior of the strong-field regime for eccentric orbits, and we have demonstrated this more explicitly through Eq. (65).

Using the conjecture (66), the formula (74) can be rewritten as

$$C(\delta_{(0)}) = -\frac{1-e_\star}{e_\star} + \frac{8\sqrt{2}(-3+e_\star)^2\sqrt{3+e_\star}\,f_{(1)}^\phi(p_\star,e_\star,0)}{e_\star\sqrt{1+e_\star}\,\mathcal{A}(e_\star)}\delta_{(0)}\log(\delta_{(0)}) + O(\delta_{(0)}), \qquad (75)$$

in terms of the self-force evaluated at the periapsis.

**Apoapsis, periapsis and radius**   The apoapsis $r_a := \frac{p}{1-e}$ slowly decreases as we approach the separatrix. More precisely, the solution (65) provides

$$r_a(\delta_{(0)}) = \frac{(6+2e_\star)}{1-e_\star} + \frac{\big(8\mathcal{G}_e(e_\star)+(1-e_\star)\mathcal{G}_\delta(e_\star)\big)}{2(e_\star-1)^2 e_\star \mathcal{A}(e_\star)}\delta_{(0)}^2\log(\delta_{(0)}) + O(\delta_{(0)}^2). \qquad (76)$$

This can be compared with the periapsis $r_p := \frac{p}{1+e}$ which instead behaves as

$$r_p(\delta_{(0)}) = \frac{(6+2e_\star)}{1+e_\star} + \frac{(3+e_\star)}{2(1+e_\star)^2}\delta_{(0)} - \frac{\big(8\mathcal{G}_e(e_\star)+(1-e_\star)\mathcal{G}_\delta(e_\star)\big)}{4e_\star(1+e_\star)^2\mathcal{A}(e_\star)}\delta_{(0)}^2\log(\delta_{(0)}) + O(\delta_{(0)}^2). \qquad (77)$$

The distance between the two roots of the radial potential grows during the late inspiral. This explains the increase in eccentricity in the approach to the separatrix.

The late-time evolution of the radius combines two distinct motions: fast oscillations between the two roots $r_a$ and $r_p$ and slow changes due to the gradual decrease of $\delta_{(0)}$ and increase of the eccentricity.

The adiabatic evolution (65), (63) and (69) together with Eq. (64) allow to compute the 0PA expression of $r(\delta_{(0)})$ which admits the expansion

$$r_{(0)}(\delta_{(0)}) = \frac{p_\star}{1+e_\star\cos\psi_r^\star} + \frac{(3+e_\star)(1+\cos(\psi_{r\star}))}{4\big(1+e_\star\cos(\psi_{r\star})\big)^2}\delta_{(0)} + O(\delta_{(0)}^2\log(\delta_{(0)})). \qquad (78)$$

The function $p_{(0)}(\delta_{(0)})/(1+e_{(0)}(\delta_{(0)})\cos\psi_{r(0)}(\delta_{(0)}))$ where we substitute the leading behavior of $e_{(0)}$, $p_{(0)}$ and $\psi_{r(0)}$ is displayed on Fig. 3. However, as already noted before, the 0PA order inaccurately captures the evolution of the radius near the separatrix. Higher PA order will correct this behavior with even more leading terms. Instead, a transition to plunge regime need to take place, which is not described here. In a more complete treatment, we expect to reproduce the qualitative feature that the relativistic anomaly at the separatrix $\psi_{r\star}$ be reached from above or below depending upon its value at the separatrix, as seen in the asymptotic expansion (34) and in phenomenological models [22, 25].

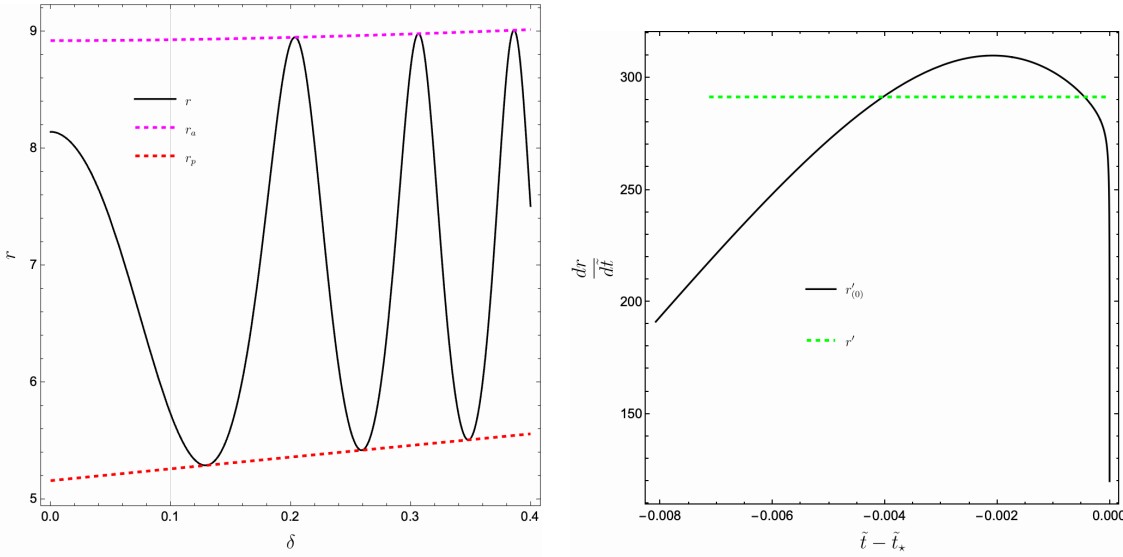

Figure 3: Left panel: evolution of the 0PA radius $r_{(0)}(\delta_{(0)}(\tilde{t}))$(solid line) for small values of $\tilde{t}_\star - \tilde{t}$. The earliest time depicted $\tilde{t}_\star - \tilde{t} \approx -0.1213$ corresponds to $\delta_{(0)} \approx 0.4$. The anomalous phase at the separatrix has been chosen to be $\psi_{r\star} = 2.4$ and the final eccentricity is $e_\star = 0.2672$. The dashed colored lines indicate the instantaneous locations of the periapsis and apoapsis, which evolve at orders $O(\delta_{(0)}(t))$ and $O(\delta_{(0)}^2(t))$, respectively. Finally, the dashed green line represents the leading behavior of the radius in time domain, as it is described in eq. (34). Here, the mass ratio is $\varepsilon = 10^{-4}$. As we can see, the secondary performs a few orbital cycles before reaching the separatrix. Given the choice of parameters $\psi_{r\star}$ and $e_\star$ the ending value of $r$ is 8.138.
Right panel: comparison of the derivatives with respect to $\tilde{t}$ of the leading behavior of the radius (dashed green line) with its 0PA counterpart (black line). As the self-force contributing terms vanish at the leading order of $r$ (see eq. (34)), the late evolution of the radius is geodesic and its derivative is smooth at the separatrix. On the opposite, the leading order of $\frac{dr_{(0)}}{d\tilde{t}}$ diverges. This exhibits the inability of the 0PA expansion to accurately capture the evolution of r at the separatrix.

**Redshift**   Expanding Eq. (2) we obtain

$$\frac{dt}{d\tau} = \gamma_\star - \frac{(3+e_\star)\cos^2\left(\frac{\psi_{r\star}}{2}\right)}{\sqrt{2}\sqrt{9-e_\star^2}\left(2+e_\star-e_\star\cos(\psi_{r\star})\right)^2}\delta_{(0)} + O(\delta_{(0)}^2\log(\delta_{(0)})), \tag{79}$$

where $\gamma_\star := \frac{E_\star}{1-\frac{2}{r_\star}} = \frac{2\sqrt{2}(3+e_\star)}{\sqrt{9-e_\star^2}\left(2+e_\star-e_\star\cos(\psi_{r\star})\right)}$ is the redshift at the separatrix. Using the explicit solution (63) we obtain

$$t = t_\star - \gamma_\star(\tau_\star - \tau) + \frac{\gamma_\star^{5/2}(1+\cos(\psi_{r\star}))(2L_1+\log(-L_1))\sqrt{\mathcal{A}(e_\star)}\sqrt{e_\star(3-e_\star)}}{24\sqrt{2}e_\star\sqrt{3+e_\star}(-L_1)^{3/2}}(\tau_\star - \tau)^{3/2}, \tag{80}$$

where we have defined

$$L_1 := \log\left(\frac{4e^{-1+\frac{2\mathcal{B}(e_\star)}{e_\star}}\mathcal{A}(e_\star)}{e_\star}\gamma_\star(\tau_\star - \tau)\right). \tag{81}$$

**Energy and angular momentum**  Combining this result with Eqs. (70), (71) and (64) allows to write the energy and angular momentum as a function of $\tau$

$$E(\tau) = E_\star(e_\star) + (\tau_\star - \tau) \tag{82}$$
$$\times \left\{ -\frac{(8\mathcal{G}_e(e_\star) + (1-e_\star)\mathcal{G}_\delta(e_\star))\gamma_\star}{2\sqrt{2}e_\star(9-e_\star^2)^{3/2}} - \frac{\gamma_\star}{8\sqrt{2}e_\star^2(1+e_\star)(9-e_\star^2)^{3/2}\log(\tau_\star - \tau)} \right.$$
$$\times \left[ e_\star\left(3 - 6e_\star + e_\star^2 + 2e_\star^3\right)\mathcal{A}(e_\star) - 8(1+e_\star) \right.$$
$$\left. \times \left( -8e_\star\mathcal{F}_e(e_\star) + (e_\star - 1)e_\star\mathcal{F}_\delta(e_\star) + 2\mathcal{B}(e_\star)\left(8\mathcal{G}_e(e_\star) + (1-e_\star)\mathcal{G}_\delta(e_\star)\right) \right) \right] \right\}$$
$$+ O\left( \frac{(\tau_\star - \tau)}{\log^2(\tau_\star - \tau)} \right),$$

$$L(\tau) = L_\star(e_\star) + (\tau_\star - \tau) \tag{83}$$
$$\times \left\{ -\frac{(8\mathcal{G}_e(e_\star) + (1-e_\star)\mathcal{G}_\delta(e_\star))\gamma_\star}{e_\star\left(3 + 2e_\star - e_\star^2\right)^{3/2}} - \frac{\gamma_\star}{4e_\star^2\left(3 + 2e_\star - e_\star^2\right)^{3/2}\log(\tau_\star - \tau)} \right.$$
$$\times \left[ \left[ e_\star(3 + e_\star^2)\mathcal{A}(e_\star) - 8\left( -8e_\star\mathcal{F}_e(e_\star) + (e_\star - 1)e_\star\mathcal{F}_\delta(e_\star) \right.\right.\right.$$
$$\left.\left.\left. + 2\mathcal{B}(e_\star)\left(8\mathcal{G}_e(e_\star) + \mathcal{G}_\delta(e_\star) - e_\star\mathcal{G}_\delta(e_\star)\right) \right) \right] \right] \right\} + O\left( \frac{(\tau - \tau_\star)}{\log^2(\tau_\star - \tau)} \right).$$

**Motion in phase space**  The motions in the $(e, p)$ and $(E, L)$ phase spaces display a strong contrast. In the $(e, p)$ phase space, Eq. (74) implies that the approach to the separatrix is along a line that crosses the separatrix line $p_\star = 6 + 2e_\star$. Instead in the $(E, L)$ phase space, Eq. (72) implies that the approach to the separatrix becomes asymptotically tangential to the the separatrix line $(E_\star(e_\star), L_\star(e_\star))$ given by Eqs. (7)-(8). Such a distinct behavior is clearly displayed on Figure 4.

# 4  Conclusion and outlook

We derived analytically the solution to the adiabatic equations for the forced geodesic motion with eccentricity around the Schwarzschild black hole in the approach to the separatrix, in terms of self-force coefficients evaluated at the separatrix. We found that the explicit solution is expressed in terms of Lambert $W_{-1}$ function, which appears as a key mathematical feature of this dynamical system.

It is known since the work [43] that the evolution in the phase space $(p, e)$ is such that the separatrix line $p = 6 + 2e$ is crossed head on. Here, we obtained analytically, after solving for the motion at subleading order in the deviation from the separatrix, that the evolution in the phase space $(E, L)$ is such that the separatrix line is crossed asymptotically tangentially, see Figure 4. Since the energy and angular momentum are well-defined beyond the separatrix, this suggests to formulate a transition-to-plunge regime where $(E, L)$ are defined as perturbations of their values at the separatrix crossing. Our result in terms of the proper time expansion is

$$E = E_\star + \varepsilon \, \Omega_{\phi\star}^{(0)} \kappa_\star(\tau_\star - \tau)\left(1 + \frac{E_\star^{\log}}{\log(\tau_\star - \tau)} + O(\log^{-2}(\tau_\star - \tau))\right), \tag{84a}$$

$$L = L_\star + \varepsilon \kappa_\star(\tau_\star - \tau)\left(1 + \frac{L_\star^{\log}}{\log(\tau_\star - \tau)} + O(\log^{-2}(\tau_\star - \tau))\right), \tag{84b}$$

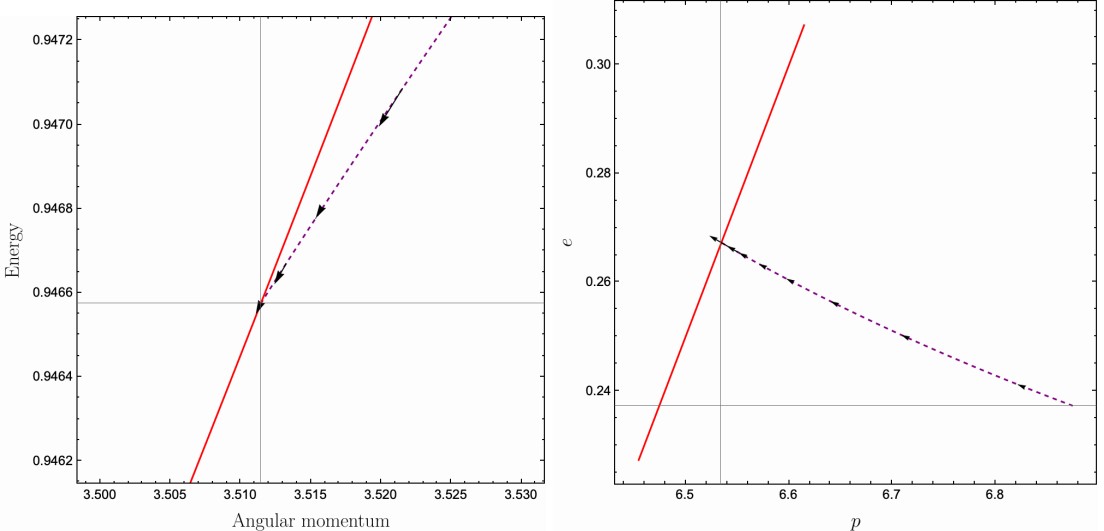

Figure 4: Late inspiral in phase space. On the left panel, the motion in the $(L, E)$ phase space (dashed line) becomes increasingly tangent to the separatrix (solid line), eventually gazing it. On the right panel, the same dynamics is shown but in the $(p, e)$ phase space. There, the secondary approaches the separatrix (solid line) head-on.

where $\Omega_{\phi\star}^{(0)}$ is given in Eq. (58) and $\kappa_\star = \kappa_\star(e_\star, \psi_{r\star})$ depends upon the relativistic anomaly $\psi_{r\star}$ at separatrix crossing through its dependency in the redshift, see Eq. (82). If we admit our conjecture (66), the leading coefficient $\kappa_\star$ is linear in the $\phi$ component of the self-force at the periapsis. The expansion (84) admits a circular limit which is compatible with the results derived in [29]. This suggests to formulate a transition-to-plunge regime based on the critical analytical solutions around the separatrix values of the energy and angular momentum [44, 45].[2] The conjecture (66) points to the role of the periapsis for the transition-to-plunge regime at an arbitrary relativistic anomaly $\psi_{r\star}$ at the separatrix.

For a generic eccentric orbit, the transition-to-plunge regime will depend upon the eccentricity $e_\star$ and the relativistic anomaly $\psi_{r\star}$ at separatrix crossing, which looses its intuitive meaning after crossing since the periapsis ceases to exist. Contrary to the quasi-circular case, the radius evolves both under the orbital and the radiation reaction timescales. A formulation using angle-action variables is therefore necessary in order to formulate a multiscale expansion scheme. We foresee that the recently obtained analytical maps between quasi-Keplerian angles and Mino time action angles [36] will be instrumental for that purpose. Any transition-to-plunge expansion scheme should reproduce at early times the late time evolution of the inspiral phase that we derived. We leave the development of a transition-to-plunge expansion scheme to further work.

## Acknowledgments

We would like to thank Leor Barack and Maarten van de Meent for interesting discussions and Maarten van de Meent for providing self-force data. G.C. is Research Director of the F.R.S.-FNRS.

---

[2]Note that critical solutions also exist in the scattering range $E > 1$ [46].

# A   Osculating equations

The osculating equations provide the equations of motion of an accelerated point particle by considering self-force effects. Following the notation introduced in Sec. 2.4, the functions that define the osculating equations for a particle orbiting around a Schwarzschild background in the equatorial plane read

$$\mathcal{F}_p(p,e,\psi_r) = \mathcal{F}_{p,\phi}(p,e,\psi_r)f^\phi(p,e,\psi_r) + \mathcal{F}_{p,r}(p,e,\psi_r)f^r(p,e,\psi_r), \tag{A.1}$$

$$\mathcal{F}_e(p,e,\psi_r) = \mathcal{F}_{e,\phi}(p,e,\psi_r)f^\phi(p,e,\psi_r) + \mathcal{F}_{e,r}(p,e,\psi_r)f^r(p,e,\psi_r), \tag{A.2}$$

$$f_r(p,e,\psi_r) = \frac{(e\cos(\psi_r)+1)^2\sqrt{-2e\cos(\psi_r)+p-6}(-2e\cos(\psi_r)+p-2)}{p^2\sqrt{(p-2)^2-4e^2}}, \tag{A.3}$$

$$\delta f_r(p,e,\psi_r) = \delta f_{r,\phi}(p,e,\psi_r)f^\phi(p,e,\psi_r) + \delta f_{r,r}(p,e,\psi_r)f^r(p,e,\psi_r), \tag{A.4}$$

$$f_\phi(p,e,\psi_r) = \frac{p(e\cos(\psi_r)+1)^2(-2e\cos(\psi_r)+p-2)}{\sqrt{p^5((p-2)^2-4e^2)}}, \tag{A.5}$$

where

$$\begin{aligned}\mathcal{F}_{p,\phi}(p,e,\psi_r) = {}& \frac{p^{3/2}\left(-3-e^2+p\right)(-2+p-2e\cos(\psi_r))}{\left(-4e^2+(-6+p)^2\right)\sqrt{-4e^2+(-2+p)^2}(1+e\cos(\psi_r))^2} \\ &\times\Big[36+6e^2-18p-e^2p+2p^2+e\left(12+3e^2-4p\right)\cos(\psi_r) \\ &\quad -e^2(-6+p)\cos(2\psi_r)+e^3\cos(3\psi_r)\Big],\end{aligned} \tag{A.6}$$

$$\mathcal{F}_{p,r}(p,e,\psi_r) = -\frac{\left(2e\left(3+e^2-p\right)p\sqrt{-6+p-2e\cos(\psi_r)}(2-p+2e\cos(\psi_r))\sin(\psi_r)\right)}{\left(-4e^2+(-6+p)^2\right)\sqrt{-4e^2+(-2+p)^2}}, \tag{A.7}$$

and

$$\begin{aligned}\mathcal{F}_{e,\phi}(p,e,\psi_r) = {}& \frac{p\left(-e^2+p-3\right)(-2e\cos(\psi_r)+p-2)}{2\left((p-6)^2-4e^2\right)\sqrt{p((p-2)^2-4e^2)}(e\cos(\psi_r)+1)^2} \\ &\times\Big[e\big(e\left(2e^2-p+6\right)\cos(3\psi_r)-(p-6)\left(2e^2-p+6\right)\cos(2\psi_r) \\ &\quad -2e^2p+20e^2+3p^2-32p+60\big) \\ &\quad +\left(6e^4+e^2(42-11p)+4\left(p^2-9p+18\right)\right)\cos(\psi_r)\Big],\end{aligned} \tag{A.8}$$

$$\mathcal{F}_{e,r}(p,e,\psi_r) = \frac{\left(e^2-p+3\right)\left(2e^2-p+6\right)\sin(\psi_r)\sqrt{-2e\cos(\psi_r)+p-6}(-2e\cos(\psi_r)+p-2)}{\left((p-6)^2-4e^2\right)\sqrt{(p-2)^2-4e^2}}. \tag{A.9}$$

Moreover

$$\begin{aligned}\delta f_{r,\phi} = {}& \frac{p\left(-e^2+p-3\right)(-2e\cos(\psi_r)+p-2)}{e\left(4e^2-(p-6)^2\right)\sqrt{p((p-2)^2-4e^2)}(e\cos(\psi_r)+1)^2} \\ &\times\sin(\psi_r)\Big[-e\left(4e^2-(p-6)^2\right)\cos(\psi_r)-(p-6)\left(e^2\cos(2\psi_r)+e^2-2p+6\right)\Big],\end{aligned} \tag{A.10}$$

$$\delta f_{r,r} = -\frac{\left(e^2-p+3\right)\sqrt{-2e\cos(\psi_r)+p-6}(2e\cos(\psi_r)-p+2)(2e+(p-6)\cos(\psi_r))}{e\left(4e^2-(p-6)^2\right)\sqrt{(p-2)^2-4e^2}}. \tag{A.11}$$

# B  Limits of elliptic integrals

In this appendix, we provide a summary of the asymptotic limits of elliptic integrals that are required to analytically describe the approach to the separatrix. The method of derivation of the asymptotic expressions of these functions given in Eq. (67) is described.

## B.1  Conventions

In this work, several elliptic integrals are used. First, using the conventions of Mathematica, the incomplete integrals of the first and second kind are respectively

$$F(\phi, m) := \int_0^{\phi} \frac{dx}{\sqrt{1 - m \sin^2(x)}}, \tag{B.1}$$

and

$$E(\phi, m) := \int_0^{\phi} \sqrt{1 - m \sin^2(x)} \, dx. \tag{B.2}$$

In particular, we define the complete integrals of the second kind by

$$K(m) := F\left(\frac{\pi}{2}, m\right), \qquad E(m) := E\left(\frac{\pi}{2}, m\right). \tag{B.3}$$

The complete integrals of the third kind $\Pi(n|m)$ have an integral representation as

$$\Pi(n, m) := \int_0^{\frac{\pi}{2}} \frac{dx}{(1 - n \sin^2(x))\sqrt{1 - m \sin^2(x)}}. \tag{B.4}$$

## B.2  Reformulation of some elliptic functions

In this section, we focus on the asymptotic behaviour of $\Pi(n|m)$ in the special case for which $n \to 1$ and $m \to 1$ simultaneously. In textbooks of mathematical functions, the asymptotic series of this function are only known when either $m \neq 1$ is fixed and $n \mapsto 1$, or, when $n \neq 1$ is fixed and $m \mapsto 1$. Here, we take advantage of the fact that $n$ and $m$ are functions of $\delta$ and are equal to 1 when $\delta = 0$. This means, that we should be able to express $\Pi(n(\delta)|m(\delta))$ as a pure function of $\delta$ (that we call $\bar{\Pi}(\delta)$) whose asymptotic expansion for $\delta \to 0$ is feasible.

For that purpose, recall that $\Pi(n|m)$ is defined as being the solution of the two partial differential equations

$$\begin{aligned}
\frac{\partial \Pi}{\partial m} &= \frac{1}{6(2 + n - 7m)}\left(3\Pi + 4\frac{\partial^2 \Pi}{\partial m^2}(n(2 - 6m) + m(11m - 7)) - 8\frac{\partial^3 \Pi}{\partial m^3}m(n - m)(m - 1)\right), \\
\frac{\partial \Pi}{\partial n} &= \frac{1}{16n - 4(m + 1)}\left(-2\Pi + \frac{\partial^2 \Pi}{\partial n^2}(-13n^2 - 3m + 8n(m + 1)) - 2\frac{\partial^3 \Pi}{\partial n^3}n(n - 1)(n - m)\right).
\end{aligned} \tag{B.5}$$

Considering that the function $\bar{\Pi}$ depends on $\delta$ via $m(\delta)$ and $n(\delta)$, we can compute its derivative with respect to $\delta$. Thanks to the chain rule

$$\frac{d\bar{\Pi}}{d\delta} = \frac{\partial \Pi}{\partial m}\frac{dm}{d\delta} + \frac{\partial \Pi}{\partial n}\frac{dn}{d\delta}, \tag{B.6}$$

we can express $\frac{\partial \Pi}{\partial n}$ as a function of $\frac{d\bar{\Pi}}{d\delta}$ and $\frac{\partial \Pi}{\partial m}$ which is function of its higher derivatives via (B.5). These latter can be replaced by the analytical formulae for the derivatives of $\Pi$ with respect to $m$ and $n$. This involves several elliptic $K$, $E$ and $\Pi$ functions.

By doing so, we conclude that $\bar{\Pi}(\delta)$ is the solution of a first order differential equation

$$0 = 2n(\delta)^3(-1+m(\delta))\bar{\Pi}'(\delta) + \left(K(m(\delta)) - \bar{\Pi}(\delta)\right)(-1+m(\delta))m(\delta)n'(\delta) \tag{B.7}$$

$$- \left(n(\delta)^2\left[2(-1+m(\delta)^2)\bar{\Pi}'(\delta) - \bar{\Pi}(\delta)(-1+m(\delta))\left(n'(\delta) - m'(\delta)\right) + E(m(\delta))m'(\delta)\right]\right)$$

$$+ \left(n(\delta)\left[2m(\delta)^2\bar{\Pi}'(\delta) + (K(m(\delta)) - E(m(\delta)))n'(\delta) + \left(E(m(\delta)) - \bar{\Pi}(\delta)\right)m'(\delta)\right]\right)$$

$$+ \left(m(\delta)\left[-2\bar{\Pi}'(\delta) + (E(m(\delta)) - K(m(\delta)))n'(\delta) + \bar{\Pi}(\delta)m'(\delta)\right]\right),$$

where a prime over a function denotes the derivatives with respect to $\delta$.

The solution can be found up to a constant that can be fixed by imposing the condition $\bar{\Pi}(1) = \Pi(n(1), m(1))$ for any eccentricity.

The results for the elliptic integrals of interest are

$$\Pi\left(\frac{2e(2+2e+\delta)}{(1+e)(4e+\delta)}, \frac{4e}{4e+\delta}\right) = \frac{\sqrt{(2+2e+\delta)(4e+\delta)}}{\delta\sqrt{8e^2+14e+3}}$$
$$\times\left(\Pi\left(\frac{2e(3+2e)}{(1+e)(1+4e)}, \frac{4e}{1+4e}\right) + \sqrt{8e^2+14e+3}\,\Lambda(\delta;e)\right),$$

$$\Pi\left(\frac{16e}{(4+\delta)(4e+\delta)}, \frac{4e}{4e+\delta}\right) = \frac{1}{5\delta}\sqrt{\delta + \frac{16e}{4+4e+\delta}}$$
$$\times\left(\sqrt{5 + \frac{20}{1+4e}}\,\Pi\left(\frac{16e}{5+20e}, \frac{4e}{1+4e}\right) + 5\Theta(\delta;e)\right), \tag{B.8}$$

where

$$\Lambda(\delta;e) = \int_1^\delta \lambda(x;e)\,dx, \qquad \lambda(x;e) := \frac{(1+e)K\left(\frac{4e}{4e+x}\right)}{(2+2e+x)^{3/2}\sqrt{4e+x}},$$

$$\Theta(\delta;e) = \int_1^\delta \theta(x;e)\,dx, \qquad \theta(x;e) := \frac{-(4e+x)E\left(\frac{4e}{4e+x}\right) + 2(2+2e+x)K\left(\frac{4e}{4e+x}\right)}{2\sqrt{(4e+x)(4+x)(4+4e+x)}}. \tag{B.9}$$

## B.3 Asymptotic expansions

The expansion of (B.8) when $\delta \to 0$ can be computed at any order, as long as we use another trick at the level of the functions $\Lambda(\delta;e)$ and $\Theta(\delta;e)$. This is necessary since the functions $\lambda$ and $\theta$ diverge at $x = 0$, but can still be integrated from 1 to 0 (this is a property we find, for instance, for the function $\log(x)$).

Let us call $\xi(x)$ a generic function of $x$ which diverges as $x \to 0$ but whose integral on the domain $[0;1]$ exists in this limit. The derivatives of this function diverge as well, but $x^n\xi^{(n)}$ admits a well defined limit as $x \to 0$ for any integer $n > 0$, where $\xi^{(n)}$ is the $n^{th}$ derivative of $\xi$. Also, $\xi$ must be such that $\frac{d^n}{dx^n}\left(x^{n+1}\xi^{(n+1)}\right)$ is well defined when $x \to 0$. The functions $x^n\xi^{(n)}(x)$ can be integrated on the range $[0;1]$ too. These properties are the same than those carried by $\lambda(x;e)$ and $\theta(x;e)$.

The function $\xi(x)$ and its derivative admit asymptotic expansions around $x = 0$ which can be reduced to

$$\xi(x) = \zeta_0 + \zeta_1\log(x) + [\zeta_2 + \zeta_3\log(x)]x + O(x^2\log x),$$
$$\xi'(x) = \frac{\hat{\zeta}_{-1}}{x} + \hat{\zeta}_0 + \hat{\zeta}_1\log(x) + \left[\hat{\zeta}_2 + \hat{\zeta}_3\log(x)\right]x + O(x^2\log x). \tag{B.10}$$

We want to develop the expansion of $\mathcal{I}(\delta) := \int_1^\delta \xi(x)\,dx$ for $\delta \to 0$ up to any order in $\delta$. For that purpose, integrating by part $m$ times allows to compute the order $\delta^m$. For example, if we

do not integrate by part, we obtain the leading order of $\mathcal{I}$ via

$$\mathcal{I}(\delta) = \mathcal{I}_1 + O(\delta), \tag{B.11}$$

where

$$\mathcal{I}_1 = \int_1^0 \xi(x)\,dx. \tag{B.12}$$

The next order of $\mathcal{I}$ is obtained by writing it as $\mathcal{I}(\delta) = \int_1^\delta \left(\frac{d}{dx}[x\,\xi(x)] - x\,\xi'(x)\right) dx$. This leads us to the expansion

$$\mathcal{I}(\delta) = \mathcal{I}_1 + \mathcal{I}_2(\delta)\,\delta + O(\delta^2), \tag{B.13}$$

where $\mathcal{I}_2(\delta) = \zeta_0 - \hat{\zeta}_{-1} + \zeta_1 \log(\delta)$.

The next order of $\mathcal{I}$ can be obtained as well. It is based on the identity

$$\xi(x) = -\frac{1}{2}\frac{d^2}{dx^2}\left[x^2\,\xi(x)\right] + \frac{d}{dx}\left[2x\,\xi(x)\right] + \frac{x^2}{2}\xi''(x). \tag{B.14}$$

With this in mind, we can calculate $\mathcal{I}$ and obtain

$$\mathcal{I}(\delta) = \mathcal{I}_1 + \mathcal{I}_2(\delta)\,\delta + \mathcal{I}_3(\delta)\,\delta^2 + O(\delta^3), \tag{B.15}$$

where $\mathcal{I}_3(\delta) = (\zeta_2 + \zeta_3 \log(\delta)) - \frac{1}{2}\left(\hat{\zeta}_0 + \hat{\zeta}_1 \log(\delta)\right) + \lim_{x\to 0}\left(\frac{d}{dx}\left[\frac{x^2}{4}\xi''(x)\right]\right)$. Similarly, and using identities such as (B.14) but with higher derivatives, we can manage to develop the higher order terms of the function $\mathcal{I}(\delta)$.

Applying this strategy to (B.9) allows us to write

$$\Lambda(\delta;e) = \Lambda(e) + \frac{1 + \log(64\,e) - \log(\delta)}{8\sqrt{2}\,\sqrt{e(1+e)}}\,\delta + O(\delta^2),$$

$$\Theta(\delta;e) = \Theta(e) + \frac{1 - e + (1+e)(\log(64\,e) - \log(\delta))}{8\sqrt{e(1+e)}}\,\delta + O(\delta^2), \tag{B.16}$$

where

$$\Lambda(e) = \int_1^0 \lambda(x;e)\,dx,$$

$$\Theta(e) = \int_1^0 \theta(x;e)\,dx. \tag{B.17}$$

Plugging this latter series expansions into eq. (B.8), we reproduce the expansions (53).

## C   Coefficients $\mathcal{B}$ and $\mathcal{A}$

The coefficient $\mathcal{B}(e)$ reads as

$$\begin{aligned}
\mathcal{B}(e) = {} & \frac{4e^2}{(e-1)(3+e)} - e\log(64e) + \frac{2\sqrt{2}\,e^{3/2}\sqrt{1+e}\left(11 - 3e^2\right)}{(3+e)^2(-1+e^2)}\Lambda(e) \\
& + \frac{\left(e(1+e)\right)^{3/2}}{(3+e)^2}\Theta(e) + \frac{e}{5(3+e)^2\sqrt{(1+e)(3+2e)(1+4e)}(-1+e^2)} \\
& \times \left\{110\sqrt{2}\,\sqrt{e}(1+e)\Pi\left(\frac{2e(3+2e)}{(1+e)(1+4e)}, \frac{4e}{1+4e}\right)\right. \\
& + \sqrt{5}\,\sqrt{e(3+2e)(5+4e)}\left(-1 - 2e + 2e^3 + e^4\right)\Pi\left(\frac{16e}{5+20e}, \frac{4e}{1+4e}\right) \\
& \left. - 30\sqrt{2}(1+e)e^{5/2}\Pi\left(\frac{2e(3+2e)}{1+5e+4e^2}, \frac{4e}{1+4e}\right)\right\}.
\end{aligned} \tag{C.1}$$

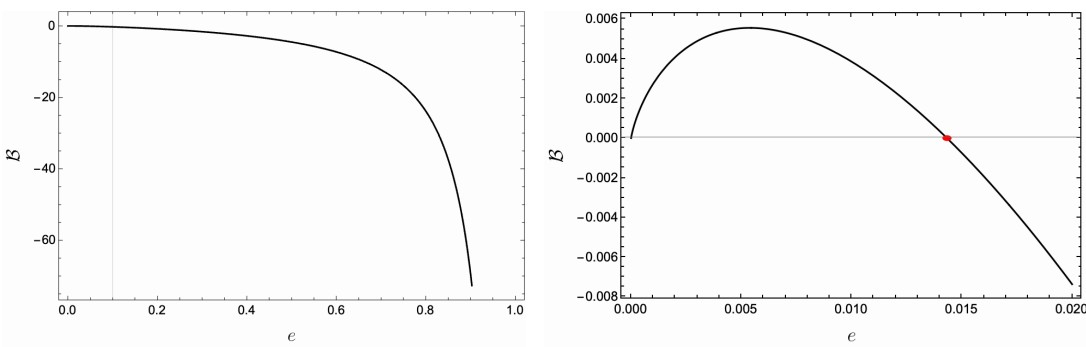

Figure 5: Left: Coefficient $\mathcal{B}(e)$ as a function of $e$. Its roots are at $e = 0$ and $e \approx 0.01433$. We have $\lim_{e \to 1} \mathcal{B}(e) = -\infty$. This coefficient is positive for small non-zero eccentricities.

The function is plotted in Fig. 5.

The term of coefficient $\mathcal{A}$ proportional to the modes of the self-force, from modes $k = -2$ to $k = 2$, read

$$\mathcal{A}_{\phi(0)}(e) = \frac{4(-3+e)\sqrt{e}\sqrt{(1+e)(3+e)}}{3(1-e)^{7/2}}\left[\sqrt{2e}\sqrt{1-e}\left(51+2e+19e^2-12e^3\right)\right.$$
$$\left. -3(1+e)^2\left(-7-7e+4e^2\right)\arcsin\left(\sqrt{2}\sqrt{\frac{e}{1+e}}\right)\right],$$

$$\mathcal{A}_{\phi(1)}(e) = \frac{4(-3+e)\sqrt{(1+e)(3+e)}}{3\sqrt{e}(1-e)^{7/2}}\left[\sqrt{2e}\sqrt{1-e}\left(3-20e-73e^2+30e^3\right)\right. \quad \text{(C.2)}$$
$$\left. +3(1+e)^2\left(-1+e-18e^2+8e^3\right)\arcsin\left(\sqrt{2}\sqrt{\frac{e}{1+e}}\right)\right]$$
$$= \mathcal{A}_{\phi(-1)}(e),$$

$$\mathcal{A}_{\phi(2)}(e) = \frac{4(-3+e)\sqrt{(1+e)(3+e)}}{3(1-e)^{7/2}e^{3/2}}$$
$$\times\left[\sqrt{2e}\sqrt{1-e}\left(-114+124e+203e^2-194e^3-19e^4+60e^5\right)\right.$$
$$\left. +3(1+e)^2(38-130e+153e^2-55e^3+4e^4)\arcsin\left(\sqrt{2}\sqrt{\frac{e}{1+e}}\right)\right]$$
$$= \mathcal{A}_{\phi(-2)}(e).$$

The coefficients $\mathcal{A}_{r(k)}$ can be written in a general way, for any $k$. We have

$$\mathcal{A}_{r(k)} = 2e^{3/2}(e-3)(1+e)^{5/2}\left(\mathcal{K}(e;-k)-\mathcal{K}(e;k)\right) = -\mathcal{A}_{r(-k)}. \quad \text{(C.3)}$$

# D  Subleading orders of the asymptotic adiabatic equations and solutions

This appendix provides the subleading orders of the asymptotic equations governing the evolution of the slow variables and their asymptotic solutions.

The asymptotic adiabatic equations that drive the evolution of $\delta_{(0)}$ and $e_{(0)}$ read as

$$\frac{d\delta_{(0)}}{d\tilde{t}} = \frac{\mathcal{A}(e_\star)}{\delta_{(0)}\big(e_\star \log(\delta_{(0)}) + \mathcal{B}(e_\star)\big)} + \frac{1}{\big(e_\star \log(\delta_{(0)}) + \mathcal{B}(e_\star)\big)^2} \tag{D.1a}$$

$$\times \left\{ \mathcal{E}_\delta(e_\star) + \frac{e_\star - 1}{8}\big[\mathcal{B}(e_\star)\mathcal{A}'(e_\star) - \mathcal{A}(e_\star)\mathcal{B}'(e_\star)\big] \right.$$

$$\left. + \left[\frac{e_\star - 1}{8}\big(e_\star \mathcal{A}'(e_\star) - \mathcal{A}(e_\star)\big) + \mathcal{F}_\delta(e_\star)\right]\log(\delta_{(0)}) + \mathcal{G}_\delta(e_\star)\log^2(\delta_{(0)}) \right\} + o(\delta_{(0)}^0),$$

$$\frac{de_{(0)}}{d\tilde{t}} = \frac{e_\star - 1}{8}\frac{\mathcal{A}(e_\star)}{\delta_{(0)}\big(e_\star \log(\delta_{(0)}) + \mathcal{B}(e_\star)\big)} + \frac{1}{\big(e_\star \log(\delta_{(0)}) + \mathcal{B}(e_\star)\big)^2} \tag{D.1b}$$

$$\times \left\{ \mathcal{E}_e(e_\star) + \frac{e_\star - 1}{64}\Big[(e_\star - 1)\mathcal{B}(e_\star)\mathcal{A}'(e_\star) + \mathcal{A}(e_\star)\big(\mathcal{B}(e_\star) - (e_\star - 1)\mathcal{B}'(e_\star)\big)\Big] \right.$$

$$\left. + \left[\mathcal{F}_e(e_\star) + \frac{e_\star - 1}{64}\big(\mathcal{A}(e_\star) + (e_\star - 1)e_\star \mathcal{A}'(e_\star)\big)\right]\log(\delta_{(0)}) + \mathcal{G}_e(e_\star)\log^2(\delta_{(0)}) \right\} + o(\delta_{(0)}^0).$$

The constants $\{\mathcal{C}_i\} := \{\mathcal{A}, \mathcal{E}_\delta, \mathcal{E}_e, \mathcal{F}_\delta, \mathcal{F}_e, \mathcal{G}_\delta, \mathcal{G}_e\}$ are written as $\mathcal{C}_i = \sum_k \mathcal{C}_{i,k}$ where $\mathcal{C}_{i,k}$ are real functions of $e_\star$ and of the values of the $k$ Fourier modes of the self-force at the separatrix. The explicit expressions for these constants are available on this Github repository. The equations (D.1) have been expanded up to the first subleading order in $\delta_{(0)}$. This makes it possible to derive the expression of $\frac{de_{(0)}}{d\delta_{(0)}}$ up to subleading order:

$$\frac{de_{(0)}}{d\delta_{(0)}} = \frac{e_\star - 1}{8} + \frac{\delta_{(0)}}{64\,\mathcal{A}(e_\star)\big(e_\star \log(\delta_{(0)}) + \mathcal{B}(e_\star)\big)^2}$$

$$\times \left\{ (e_\star - 1)\mathcal{A}(e_\star)\mathcal{B}(e_\star) + 64\left(\mathcal{E}_e(e_\star) - \frac{e_\star - 1}{8}\mathcal{E}_\delta(e_\star)\right) \right.$$

$$+ \left[(e_\star - 1)e_\star \mathcal{A}(e_\star) + 64\left(\mathcal{F}_e(e_\star) - \frac{e_\star - 1}{8}\mathcal{F}_\delta(e_\star)\right)\right]\log(\delta_{(0)}) \tag{D.2}$$

$$\left. + 64\left[\mathcal{G}_e(e_\star) - \frac{e_\star - 1}{8}\mathcal{G}_\delta(e_\star)\right]\log^2(\delta_{(0)}) \right\} + o(\delta_{(0)}).$$

The solution to this equation is

$$e_{(0)}(\delta_{(0)}) = e_\star + \frac{e_\star - 1}{8}\delta_{(0)} + \frac{e_\star - 1}{128}\delta_{(0)}^2 - \frac{1}{32\,e_\star^3\,\mathcal{A}(e_\star)}\left\{ e_\star \delta_{(0)}^2\left[ -16\,e_\star \mathcal{F}_e(e_\star) \right.\right.$$

$$+ 2(e_\star - 1)e_\star \mathcal{F}_\delta(e_\star) + (2e_\star \log(\delta_{(0)}) - e_\star - 2\mathcal{B}(e_\star))(-8\,\mathcal{G}_e(e_\star) + (e_\star - 1)\mathcal{G}_\delta(e_\star))\Big] \tag{D.3}$$

$$+ 4e^{-\frac{2\mathcal{B}(e_\star)}{e_\star}} Ei\left[2\left(\log(\delta_{(0)}) + \frac{\mathcal{B}(e_\star)}{e_\star}\right)\right]\left[ -8\,e_\star^2 \mathcal{E}_e(e_\star) + (e_\star - 1)e_\star^2 \mathcal{E}_\delta(e_\star) \right.$$

$$\left.\left. + \mathcal{B}(e_\star)\big(8\,e_\star \mathcal{F}_e(e_\star) - (e_\star - 1)e_\star \mathcal{F}_\delta(e_\star) + \mathcal{B}(e_\star)(-8\,\mathcal{G}_e(e_\star) + (e_\star - 1)\mathcal{G}_\delta(e_\star))\big)\right]\right\},$$

where $Ei$ is the exponential integral function defined as $Ei(z) = \int_{-\infty}^z \frac{e^t}{t}\,dt$. This asymptotic solution provides the subleading orders of Eq. (65) given in the body of the paper. In particular, we see from this expression that the residual of the leading solution (65) is $O(\delta_{(0)}^2 \log(\delta_{(0)}))$.

## E  Lambert W function

This section provides elements on the Lambert $W$ function that are useful for the main text. In this section, the symbol $e$ always denotes Napier's constant, not the eccentricity.

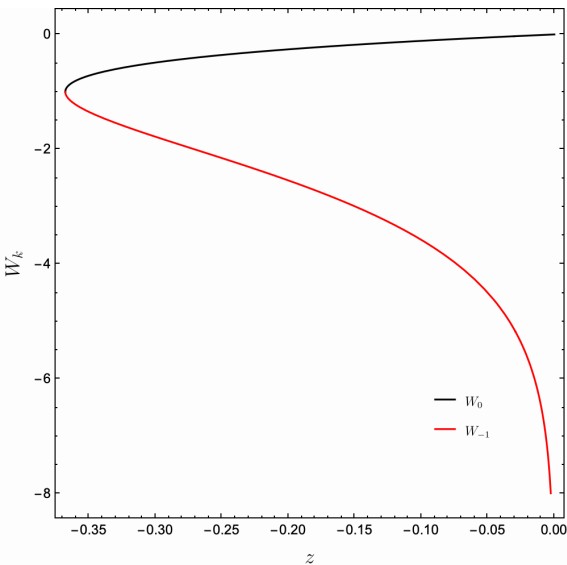

Figure 6: The two real branches of the $W$ Lambert function in the domain $z \in [-\frac{1}{e}, 0[$. In black, $W_0(z)$ has the image set $[-1, 0[$. In red, $W_{-1}(z)$ has the image set $[-1, -\infty[$. It is monotonically decreasing with $z$ and asymptotes to $-\infty$ as $z \to 0$.

The Lambert $W$ function is the multivalued inverse of the function $w \to w\, e^w$:

$$W(z) = w \quad \Longleftrightarrow \quad z = w\, e^w . \tag{E.1}$$

The distinct branches of this function are denoted with an integer $k$. Among these branches, two are real, the branches $k = 0$ and $k = -1$, which interest us most. On the one hand, the branch $k = 0$ provides real values of the Lambert function for any $z \geq -e^{-1}$. On the other hand, the real branch $k = -1$ is defined for $-e^{-1} \leq z < 0$ and can be seen as an extension of the branch 0 on the real axis. The plots of the two real branches of the $W$ function are given in Figure 6.

In particular, we have the identities

$$
\begin{aligned}
W_k(-e^{-1}) &= -1 , && k = 0, -1 , \\
W_0(0) &= 0 , && \\
W_{-1}(z) &\to -\infty , && \text{as } z \to 0 , \\
W_0(e) &= 1 , && \\
e^{-W_k(z)} &= \frac{W_k(z)}{z} , && \text{if } z \neq 0 .
\end{aligned}
\tag{E.2}
$$

The $W$ function is a solution to the differential equation

$$\frac{df(z)}{dz} = \frac{1}{e^{f(z)}(1 + f(z))} . \tag{E.3}$$

The principal branch of $W$ is analytic at 0 and it can be shown [47] that the series expansion around $z = 0$ is

$$W_0(z) = \sum_{n=1}^{\infty} \frac{(-n)^{n-1}}{n!} z^n . \tag{E.4}$$

The other real branch, which is the one relevant for the late time dynamics of the inspiral, is

expanded as $z \mapsto 0$ as

$$
\begin{aligned}
W_{-1}(z) = L_1 - L_2 &+ \frac{L_2}{L_1} + \frac{L_2(-2 + L_2)}{2L_1^2} + \frac{L_2(6 - 9L_2 + 2L_2^2)}{6L_1^3} \\
&+ \frac{L_2(-12 + 36L_2 - 22L_2^2 + 3L_2^3)}{12L_1^4} + O\left(\left(\frac{L_2}{L_1}\right)^5\right),
\end{aligned}
\tag{E.5}
$$

where $L_1 := \log(-z)$ and $L_2 := \log(-\log(-z))$. This expansion is numerically accurate, as it can be seen on Figure 7, which compares the function $W_{-1}$ with its (truncated) approximation (E.5).

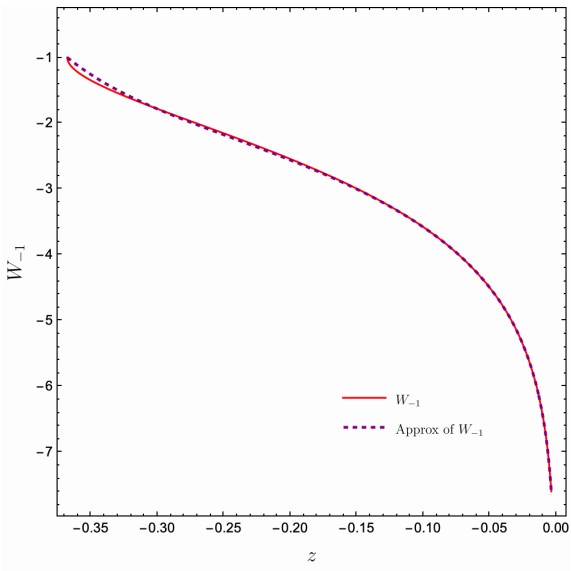

Figure 7: Solid line: the function $W_{-1}(z)$. Dashed line: the asymptotic expansion (E.5) (truncated to the order shown) of $W_{-1}(z)$.

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
