# Peer review of "Approach to the separatrix with eccentric orbits"

_SciPost Physics Core, doi:SciPost Phys. Core 8, 059 (2025)_

## Round 1 · Referee Report · Anonymous (Referee 1) · 2025-4-14

Report
This paper studies how bound orbits of a Schwarzschild black hole evolve under gravitational radiation reaction near the separatrix between bound and plunging orbits. The paper analyses the adiabatic evolution of such inspirals and forms a foundation upon which to develop future transition (from inspiral to plunge) frameworks extending what as been done for quasi-circular inspirals. The results in this paper significantly extend previous analysis (e.g., Cutler, Kennefick and Poisson and Glampedakis and Kennefick) and as such it is a very welcome addition to the literature.
I have a few minor comments and corrections that I will list below. I also have one major question that I would like the authors to answer: In Figure 3 it looks like the separatrix is reached at apoastron. I assume this cannot be a generic feature of the approach to the separatrix so has it occured in this case because of the particular parameter choice? I note that in the work by Becker and Hughes (arXiv:2410.09160) this does not appear to occur in general -- see e.g., their Fig. 4. It is also slightly confusing for the orbit to be at apastron but $\psi_{r*}=2.4$ (rather than $\pi$). Has this occurred due to the rapid evolution of $p$ and $e$?
I have a few minor comments and corrections that I will list below. I also have one major question that I would like the authors to answer: In Figure 3 it looks like the separatrix is reached at apoastron. I assume this cannot be a generic feature of the approach to the separatrix so has it occured in this case because of the particular parameter choice? I note that in the work by Becker and Hughes (arXiv:2410.09160) this does not appear to occur in general -- see e.g., their Fig. 4. It is also slightly confusing for the orbit to be at apastron but $\psi_{r*}=2.4$ (rather than $\pi$). Has this occurred due to the rapid evolution of $p$ and $e$?
Requested changes
- An explanation for the behaviour mentioned above in Fig. 3 should be provided
- In the second paragraph there is a citation to Stein and Warburton -- Ref. [28]. This citation is about the separatrix but the sentence making the citation is referring to an evolving inspiral. I suggest a change of wording here to clarify, e.g., "for a thorough description of the separatrix, see [28])
- In Eq. (2.7) and (2.8) the * notation is used for the first time. I think it should be clarified that "hereafter a * subscript denotes a quantity evaluated at the separatrix"
- In Eq. (3.1) I am not sure $e_{(0)}$ has been defined. Although it is clear what is being referred to it looks to me like its definition should appear in Eq. (2.8)
- The authors may consider merging the two plots in Fig. 1 into one plot, with $\log\delta$ plotted on the x-axis
- page 16 "as it his described" -> "as it is described"
Recommendation
Publish (surpasses expectations and criteria for this Journal; among top 10%)

Author: Guillaume Lhost on 2025-06-05 [id 5546]
(in reply to Report 1 on 2025-04-14)Dear referee,
Thank you for your request changes and questions. Here are the answers and comments to your questions.
Minor corrections: - Grammatical suggestions have been applied, - The notation $\star$ is now explained after Eq. (2.7), - The adiabatic orbital eccentricity $e_{(0)}$ is now defined after Eq. (2.36) and this definition is referred to after Eq. (3.1), - Figure 1 has been updated according to your comments, - Eq. (C.1) was simplified, - We simplified some analytical expression in the Appendix such as the coefficient $\mathcal B(e)$.
Major correction: - We thank you for pointing to the unnatural behavior of the 0PA solution close to the separatrix. We added the additional section 2.5. which obtains the asymptotic solution of the inspiral motion without a post-adiabatic expansion. This asymptotic solution admits the standard feature that the derivative of the radius with respect to time at the separatrix takes positive or negative values depending upon the relativistic anomaly at the separatrix. We then contrasted that behavior with the near-separatrix limit of the adiabatic solution, which does not admit that feature. We added further comments in Figure 3, which now contains a left and right panel. - We updated our comments at the end of Section 3.3. and after Eq. (3.30).
The paper has been updated on ArXiv and can be read in https://arxiv.org/abs/2412.04249.
Best regards, Guillaume L. and Geoffrey. C

---

## Editorial Decision

published